# Restoring Exploration after Post-Training:
# Latent Exploration Decoding for Large Reasoning Models

**Wenhui Tan** [1]   **Fiorenzo Parascandolo** [2 3]   **Enver Sangineto** [2]   **Jianzhong Ju** [4]   **Zhenbo Luo** [4]   **Qian Cao** [1]
**Rita Cucchiara** [2]   **Ruihua Song** [1]   **Jian Luan** [4]

## Abstract

Large Reasoning Models (LRMs) have recently achieved strong mathematical and code reasoning performance through Reinforcement Learning (RL) post-training. However, we show that modern reasoning post-training induces an unintended exploration collapse: temperature-based sampling no longer increases pass@$n$ accuracy. Empirically, the final-layer posterior of post-trained LRMs exhibit sharply reduced entropy, while the entropy of intermediate layers remains relatively high. Motivated by this entropy asymmetry, we propose Latent Exploration Decoding (LED), a depth-conditioned decoding strategy. LED aggregates intermediate posteriors via cumulative sum and selects depth configurations with maximal entropy as exploration candidates. Without additional training or parameters, LED consistently improves pass@1 and pass@16 accuracy by 0.61 and 1.03 percentage points across multiple reasoning benchmarks and models. Furthermore, integrating LED into reinforcement learning, e.g., using GRPO as the rollout strategy, yields faster reward improvement and higher final performance, due to the efficient exploration capability of LED. Project page: https://github.com/AlbertTan404/LED.

## 1. Introduction

Large Reasoning Models (LRMs), also referred to as reasoning Large Language Models (LLMs), have demonstrated rapid progresses on complex tasks such as mathematics, science, and coding (Jaech et al., 2024; Guo et al., 2025;

[1]Gaoling School of Artificial Intelligence, Renmin University of China, Beijing, China [2]University of Modena and Reggio Emilia, Italy [3]University of Pisa, Italy [4]MiLM Plus, Xiaomi Inc., Beijing, China. Correspondence to: Ruihua Song <rsong@ruc.edu.cn>, Jian Luan <luanjian@xiaomi.com>.

*Proceedings of the 43rd International Conference on Machine Learning*, Seoul, South Korea. PMLR 306, 2026. Copyright 2026 by the author(s).

Team et al., 2025; Yang et al., 2025; Xiao et al., 2026). This progress is largely driven by two key techniques. First, models are prompted to perform step-by-step Chain-of-Thought (CoT) reasoning within "$\langle$think$\rangle\langle$/think$\rangle$" tags, which is termed as *DeepThink* (Wei et al., 2022; Jaech et al., 2024; Guo et al., 2025), before generating responses. Second, Reinforcement Learning (RL) based post-training (Shao et al., 2024; Yu et al., 2025; Ye et al., 2025; Wang et al., 2025) aligns model outputs with correctness-oriented objectives, substantially improving pass@1 accuracy.

Despite these improvements, we observe that LRMs exhibit limited gains in pass@$n$ accuracy (i.e., for each question, a model generates at least one correct answer over $n$ attempts, $n > 1$), a metric widely used to evaluate a model's exploration capability, and directly reflects real-world use cases such as code generation and theorem proving, where multiple sampled candidates can be verified to obtain a correct outcome. (Chen, 2021; Chen et al., 2024). A commonly adopted strategy to enhance pass@$n$ is sampling temperature tuning (Zhu et al., 2024). For earlier LLMs (Yang et al., 2024; Grattafiori et al., 2024), increasing the sampling temperature reliably improves pass@$n$, indicating effective exploration. However, this property no longer holds for modern RL-post-trained LRMs (Yang et al., 2025; Xiaomi et al., 2025): in many cases, higher temperatures fail to improve pass@$n$ accuracy and may even degrade performance.

Several methods have been proposed to enhance test-time exploration capabilities of LRMs. DoLa (Chuang et al., 2023) proposes contrasting LLMs' final-layer posterior with latent posteriors from lower layers to amplify factual correctness. While not explicitly designed for test-time exploration, it implicitly reshapes the final-layer posterior for more effective sampling. SoftThinking (Zhang et al., 2025) adopts a "softer" exploration mechanism, by replacing hard one-hot token sampling with posterior-weighted embedding averaging, enabling a breadth-first-search-like parallel reasoning process. SoftThinking-Gumbel (Wu et al., 2025) further enhances exploration by injecting Gumbel-Softmax noise (Gumbel, 1954; Kool et al., 2019) into the final-layer posterior. (More related work is discussed in Appendix A).

Despite their success, a fundamental challenge remains unre-

solved for LRMs: *how to restore effective exploration when the final-layer posterior itself has collapsed* (Jiang et al., 2025; Cui et al., 2025). In this work, we first identify that RL post-training induces an unintended exploration collapse at the final-layer posterior: the final-layer posterior become highly confident with a low-entropy. On the other hand, we show that latent posteriors from intermediate layers still retain substantial uncertainty. This creates a sharp entropy asymmetry across depth. As a result, exploration potential still exists inside intermediate layers.

Motivated by the observations, we propose **Latent Exploration Decoding** (LED), a simple and training-free decoding strategy, which restores exploration by leveraging intermediate hidden states. Specifically, LED first obtains latent posteriors by directly feeding hidden states from intermediate layers to the language modeling head, known as *early exit* technique (Schuster et al., 2022). Then, a top-$k$ filtering is applied on the corresponding latent posteriors, only keeping the top-probability tokens from the final-layer posterior, to avoid decoding very-rare tokens. Third, all of these posteriors are aggregated by a final-to-latent cumulative sum, and the combination with highest entropy is selected as the **exploration posterior**. To balance exploration and exploitation, LED adaptively switches between latent exploration and standard decoding based on model confidence, and applies exploration only during the *DeepThink* phase.

Across multiple reasoning benchmarks (Cobbe et al., 2021; AMC, 2025; Lightman et al., 2023; Rein et al., 2024; Jain et al., 2024) and models (Xiaomi et al., 2025; Yang et al., 2025; Grattafiori et al., 2024; Guo et al., 2025), LED consistently improves pass@1 and pass@16 accuracy by 0.61 and 1.03 percentage points, over regular decoding and other strong baseline methods (Chuang et al., 2023; Zhang et al., 2025; Wu et al., 2025). The performance gain comes with negligible inference overhead and no extra training. Furthermore, by applying LED, high temperature reactivates effective exploration on RL post-trained LRMs.

To conclude, our contributions are threefold:

- We identify and analyze entropy collapse in RL-post-trained LRMs, and reveal the existence of latent entropy preserved in intermediate layers.

- We propose Latent Exploration Decoding (LED), a simple yet effective decoding strategy that restores exploration by leveraging latent representations.

- Extensive experiments across five models and six benchmarks demonstrate consistent accuracy improvements of LED, increasing pass@1 and pass@16 by 0.61 and 1.03 percentage points, respectively. Furthermore, integrating LED into GRPO as the rollout strategy yields faster reward improvement and higher final performance.

## 2. Motivation

Modern Large Reasoning Models (LRMs) rely heavily on Reinforcement Learning (RL) post-training, especially GRPO-styled RL training (Shao et al., 2024; Yu et al., 2025), to sharpen reasoning confidence and improve pass@1 accuracy. However, we find that such post-training induces an unintended *exploration collapse*: the final-layer posterior becomes over-concentrated, rendering traditional sampling-based exploration less effective.

### 2.1. RL Post-Training Induces Exploration Collapse

In Figure 1, we show the relationship between pass@$n$ accuracy, sampling temperature (higher temperature results in smoother distribution, explained in Appendix B.1), and the number of samples $n$, on multiple LRMs/LLMs. To quantify the effect of temperature on exploration, we introduce the **accuracy-temperature slope** $\alpha$, defined as the expected accuracy gain obtained by increasing the sampling temperature (estimated by least squares over pass@1 through pass@16).

For earlier LRMs such as QwQ-32B (Qwen, 2025) and DeepSeek-8B (DeepSeek-R1-Distill-Llama-8B) (Guo et al., 2025; Grattafiori et al., 2024), increasing the sampling temperature consistently improves pass@$n$ accuracy, yielding a steep positive $\alpha$. In contrast, for recent RL-post-trained LRMs, including MiMo-7B-RL and the Qwen3-T (Thinking) series, higher temperatures provide no benefit: the accuracy-temperature slope $\alpha$ becomes small or even negative. A particularly revealing comparison arises within the Qwen3 family. Qwen3-4B-I (Instruct) exhibits a stable and positive $\alpha$, whereas Qwen3-4B-T (Thinking), which underwent additional RL post-training for reasoning, shows *decreasing* pass@$n$ accuracy as temperature increases. **This suggests that effective exploration cannot be recovered via simply smoothing the output logits (i.e., increasing sampling temperature) for LRMs**.

We attribute this behavior to the optimization objectives of RL post-training algorithms. For instance, consider Group Relative Policy Optimization (GRPO) (Shao et al., 2024), a widely adopted algorithm in recent RL-post-training pipelines: GRPO-style objectives reward correct generations relative to incorrect ones across multiple rollouts, thereby explicitly optimizing pass@1-style correctness. This relative optimization implicitly concentrates probability mass onto a small number of dominant hypotheses, shrinking the effective sampling support. As RL post-training becomes more aggressive, this concentration effect intensifies, yielding final-layer posteriors that are highly confident yet low-entropy. We provide a mechanistic explanation of this entropy-collapse effect in Appendix B.2.

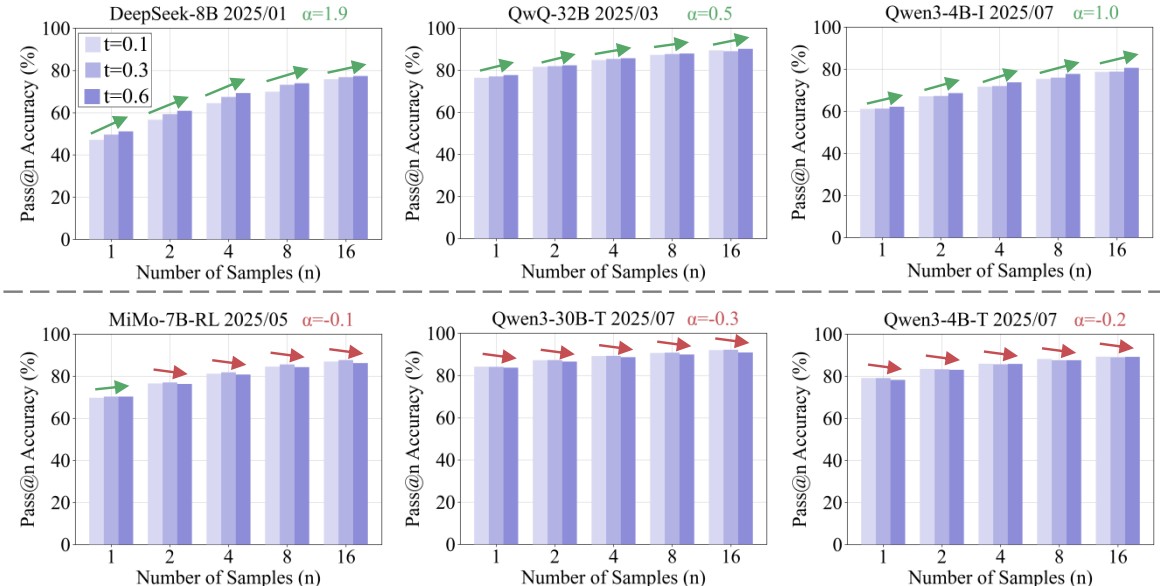

*Figure 1.* Pass@$n$ accuracy (%) for LLMs under different sampling temperatures, with darker bars representing higher values, specifically, 0.1, 0.3, and 0.6 (temperatures higher than 0.6 are not reported, as they could lead to endless looping and deteriorated performance). For earlier models or non-reasoning models, e.g., QwQ-32B, DeepSeek-8B, and Qwen3-4B-I (Instruct), higher temperature yields higher accuracy, producing a higher **accuracy-temperature slope** ($\alpha$) as noted in each subtitle. In contrast, for the latest LRMs, MiMo and Qwen3-T (Thinking) series, increasing the temperature could result in negative $\alpha$.

## 2.2. Latent Entropy Reservoirs

Prior work has shown that intermediate hidden states of LLM layers (Vaswani et al., 2017) can be directly decoded through the language modeling head, a phenomenon commonly referred to as *early exit* (Schuster et al., 2022). This is structurally consistent due to the residual connections (He et al., 2016) between Transformer (Vaswani et al., 2017) blocks. Based on this, we further analyze the decoded posteriors layer-by-layer and observe a clear trend (see Appendix B.3 for implementation details): **entropy remains high during early and intermediate layers, then decreases sharply in the final layers (Figure 2)**. This monotonic entropy decay is consistent across LLMs.

Importantly, we find that the lowest entropy is concentrated at the final layer posterior, which is directly optimized by RL post-training algorithms (Schulman et al., 2017; Rafailov et al., 2023; Shao et al., 2024). In contrast, intermediate layers retain substantial uncertainty. We refer to these intermediate layers collectively as a *latent entropy reservoir*: a region in the forward computation where the model has not yet committed to a single reasoning trajectory, and where exploration remains viable.

This provides an explanation for why temperature-based decoding becomes ineffective in recent LRMs: temperature only operates on squeezed final-layer posterior. In contrast, latent posteriors preserve exploratory semantics that can be leveraged during reasoning. This motivates exploration in latent space rather than at the final layer.

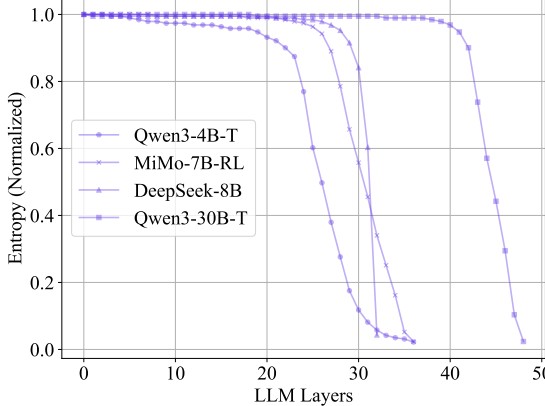

*Figure 2.* Normalized entropy across LLM layers.

## 3. Methodology

In this section, we introduce Latent Exploration Decoding (LED), a decoding strategy that leverages entropy preserved in intermediate layers of LRMs. We consider an LLM with $L$ Transformer layers. For each timestep, the model produces hidden states $\{h^1, h^2, \ldots, h^L\}$, where $h^L$ is the final-layer hidden state. We refer to layer 1 to layer $L$-1 as **latent** layers in this paper. The language modeling head (LM-Head), along with a LayerNorm module (LN) (Ba et al., 2016; Zhang & Sennrich, 2019) maps a hidden state $h^l$ to $logits^l$ over the vocabulary, followed by temperature-scaling and a softmax to produce a posterior distribution $p^l \in \mathbb{R}^V$, where $V$ is the vocabulary size.

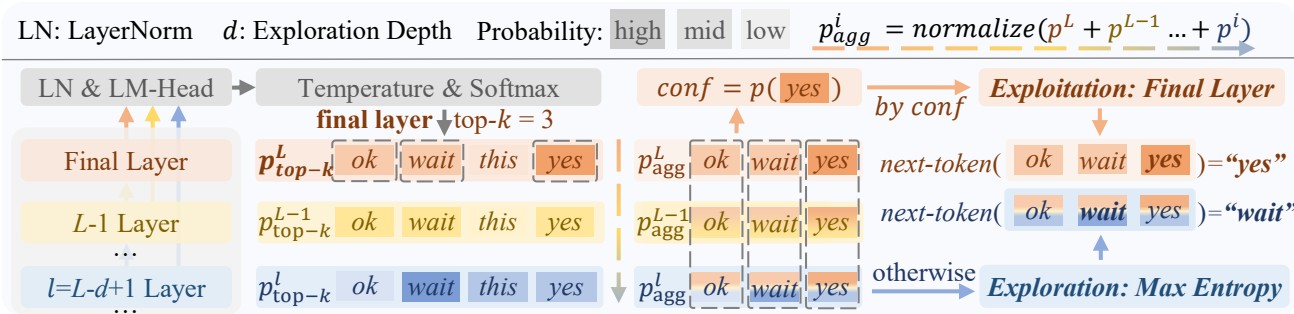

*Figure 3.* The overview of our proposed Latent Exploration Decoding (LED) method.

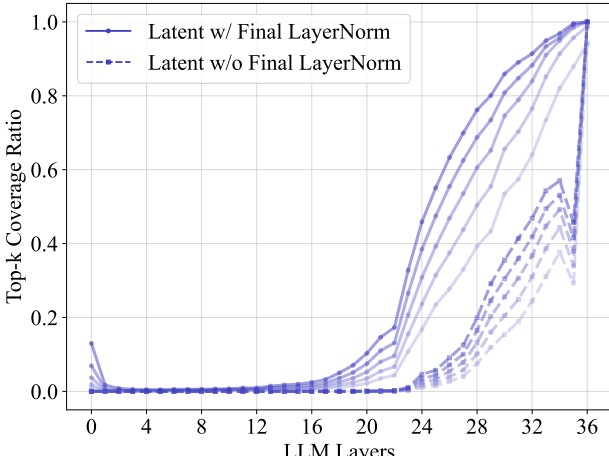

*Figure 4.* Top-$k$ coverage ratios $\{r_k^l\}_{l=1}^L$ for Qwen3-4B-Thinking ($k \in \{1, 2, 4, 8, 16\}$, averaged over all benchmarks). Darker colors correspond to greater $k$ values.

Standard decoding uses only the final posterior $p^L$ for sampling. In contrast, LED leverages a set of latent posteriors with $p^L$, obtaining $\mathbf{p} = \{p^{L-d+1}, \ldots, p^L\}$, where $d$ denotes the *Exploration Depth*. Figure 3 provides an overview of the proposed method.

### 3.1. Top-$k$ Filtering

Before introducing our decoding strategy, we first conduct a layer-wise analysis of posterior mass concentration, shown in Figure 4. Specifically, for each layer-wise posterior $\{p^l\}_{l=1}^L$, we compute the corresponding *top-$k$ coverage ratio* $\{r_k^l\}_{l=1}^L$, which measures how much probability mass of **each layer** is assigned to the **final-layer** top-$k$ candidates.

We first extract the token indices of the top-$k$ most probable tokens from the final-layer posterior:

$$\mathbf{x}_{\text{top-}k} = \{x_1, \ldots, x_k\} = \text{top-}k(p^L). \quad (1)$$

An example from Figure 3: with $k = 3$, the indices of "*ok*", "*wait*", and "*yes*" are selected, and the index of "*this*" is discarded. This procedure mirrors standard top-$k$ sampling (Fan et al., 2018; Holtzman et al., 2019), in which the

top-$k$ tokens are treated as *valid next-token candidates*.

For each layer $l \in \{L - d + 1, \ldots, L\}$, we then define the *top-$k$ filtered posterior* by restricting $p^l$ to final candidates:

$$p_{\text{top-}k}^l(i) = p^l(x_i), \quad i \in \{1, \ldots, k\}. \quad (2)$$

Finally, the top-$k$ coverage ratio for layer $l$ is computed as

$$r_k^l = \sum_{i=1}^k p^l(x_i). \quad (3)$$

As shown in Figure 4, for early layers, $r_k^l$ are nearly zero, since these layers are "immature" and contain less semantics of next-token candidates. Then latent-layer posteriors exhibit unsaturated top-$k$ coverage, which increases gradually with depth, indicating a smooth convergence process. In the end, the final-layer posterior is highly concentrated: where the top-1 coverage typically exceeds 90%, and the top-2 coverage surpasses 99%, rendering the distribution effectively near one-hot. The effect of the Final LayerNorm module is discussed in Section 4.3.

This analysis provides direct evidence of entropy collapse and posterior squeezing in modern RL-post-trained LRMs, while intermediate layers retain substantial residual uncertainty and thus serve as *entropy reservoirs*, as discussed in Section 2.2. However, latent posteriors may also assign non-negligible probability mass to non-candidate tokens, introducing noise during sampling. This observation motivates our top-$k$ filtering design: **to restrict exploration to semantically meaningful candidates, LED operates exclusively on top-$k$ filtered posteriors $p_{\text{top-}k}^l$, rather than the full-vocabulary posterior $p^l$.**

### 3.2. Cumulative Aggregation and Entropy Selection

Given filtered posteriors $\{p_{\text{top-}k}^l\}_{l=L-d+1}^L$, a key question is how to aggregate information across depth. An intuitive solution is weighted averaging, which relies on pre-defined weights, and requires hyper-parameter tuning when generalizing to different models. In contrast, we propose an

hyper-parameter free approach, which applies cumulative sum aggregation and maximum entropy selection.

Specifically, for each layer $l \in \{L - d + 1, \ldots, L\}$, we define the final-to-latent aggregated posterior as

$$p_{\text{agg}}^l(j) = \frac{\sum_{i=l}^{L} p_{\text{top-}k}^i(j)}{\sum_{j'=1}^{k} \sum_{i=l}^{L} p_{\text{top-}k}^i(j')}, \quad j \in \{1, \ldots, k\}. \quad (4)$$

This produces $d$ normalized (sum to one) candidate distributions, each corresponding to a different effective depth. This step is represented as the dashed arrow in Figure 3.

With the aggregated posteriors $\{p_{\text{agg}}^l\}_{L-d+1}^{L}$, we compute the entropy of each combination:

$$H(p_{\text{agg}}^l) = -\sum_{i=1}^{k} p_{\text{agg}}^l(i) \log p_{\text{agg}}^l(i). \quad (5)$$

The combination with the maximum entropy (the blue/third-to-last layer in Figure 3) is selected as the exploration target:

$$p_{\text{explore}} = \arg\max_l H(p_{\text{agg}}^l). \quad (6)$$

**This procedure adaptively selects the depth that provides the richest exploration signal, without manual tuning hyper-parameters.** The maximum depth $d$ could be empirically set to the layer which has negligible top-$k$ coverage ratio, where earlier layers do not introduce extra information of the candidates (discussed in Section 4.3).

### 3.3. Balancing Exploration with Exploitation

Not all tokens require exploration, as many tokens are trivially predictable. LED therefore uses a two-branch strategy.

The exploitation branch samples directly from $p_{\text{exploit}} = p_{agg}^L$ using standard decoding, and the exploration branch samples from $p_{\text{explore}}$. To decide which branch to proceed, the entropy of the final-layer posterior $H(p^L)$ is a practical criterion (Shi et al., 2025; Wang et al., 2025). However, this requires a pre-defined entropy threshold.

Considering the posterior squeeze phenomenon in LRMs (Section 2.2), we use the top-1 probability $\max_{v \in \mathcal{V}} p^L(v)$, where $\mathcal{V}$ denotes the vocabulary, as a parameter-free *confidence criterion* to decide whether to explore at certain step (the probability of token "*yes*" of the final layer in Figure 3). Higher confidence favors exploitation, while lower confidence activates exploration. **This design dynamically avoids introducing noise when predicting highly-confident or trivial tokens, balancing exploration and exploitation without extra hyper-parameter.**

### 3.4. *DeepThink*-Only Exploration

Exploration is most effective during the *DeepThink* phase, where the model actively searches over alternative reasoning paths, whereas during the final answer generation phase it is preferable to faithfully follow the established reasoning trajectory (Zhang et al., 2025). Empirically, the *DeepThink* phase accounts for the majority of generated tokens ($>$ 90%) and exhibits substantially higher entropy than the response phase. **Accordingly, LED is applied exclusively during the *DeepThink* phase, and falls back to regular sampling during response generation.**

### 3.5. Complexity Analysis

Compared to regular decoding process, LED requires storing extra hidden states with size $s$ from the last $d - 1$ layers, and mapping them to posteriors by the LM-Head ($O(ds + dV)$), computing cumulative sums over top-$k$ candidates ($O(dk)$), and calculating entropy over top-$k$ probabilities ($O(dk)$). Given that $d$ and $k$ are relative small constants (basically less than 20), our LED introduces minimal overhead. No additional model parameters or training steps are introduced in the entire pipeline.

## 4. Experiment

### 4.1. Experimental Setup

**Benchmarks.** We evaluate the proposed method across three domains using six benchmarks: (i) *Mathematics*: GSM8K (Cobbe et al., 2021), MATH-500 (Lightman et al., 2023), AIME 2024, and AIME 2025 (AMC, 2025); (ii) *Science*: GPQA-Diamond (Rein et al., 2024); and (iii) *Coding*: LiveCodeBench v5 (Jain et al., 2024).

**Metrics.** Following standard evaluation protocols (Yang et al., 2025; Guo et al., 2025), we report *Pass@1* accuracy (averaged over 16 runs) to directly measure model performance, and *Pass@16* accuracy, which estimates the model's exploration capability, by measuring whether a problem is solved correctly at least once within 16 attempts.

**Baseline Methods.** We compare our method with strong training-free baselines: (i) *CoT* (Jaech et al., 2024; Guo et al., 2025; Wei et al., 2022), which is the standard chain-of-thought reasoning; (ii) *DoLa* (Chuang et al., 2023) is a contrasting decoding (CD) (Li et al., 2023) method that contrasts hidden layers to surface factual knowledge; (iii) *SoftThinking* (ST) (Zhang et al., 2025) applies weighted sum over sampling posteriors instead of hard sampling, to perform exploration over vocabulary dimension; and (iv) *SoftThinking-Gumbel* (ST-G) (Wu et al., 2025), which further proposes adding Gumbel-Softmax noise on sampling posteriors for enhanced exploration.

**Implementation Details.** For a fair comparison, all methods use the same widely adopted sampling hyper-parameters (Yang et al., 2025): random-seed=0, temperature=0.6, top-$p$=0.95, top-$k$=20, and the maximum number of generated tokens is set to 32,768. Baseline methods are implemented using their official code and recom-

*Table 1.* Pass@1 (p@1) and pass@16 (p@16) accuracy (%) across six benchmarks and three LRMs. We bold the **best** results and underline improved performance compared to the CoT baseline. ST, ST-G, GPQA-D, and LCB denote SoftThinking, SoftThinking-Gumbel, GPQA-Diamond, and LiveCodeBench, respectively.

| | Mathmatics | | | | | | | | Science | | Coding | | Overall | |
|---|---|---|---|---|---|---|---|---|---|---|---|---|---|---|
| | AIME2024 | | AIME2025 | | GSM8K | | MATH-500 | | GPQA-D | | LCB | | | |
| | p@1 | p@16 | p@1 | p@16 | p@1 | p@16 | p@1 | p@16 | p@1 | p@16 | p@1 | p@16 | p@1 | p@16 |
| ***Qwen3-4B-Thinking*** | | | | | | | | | | | | | | |
| CoT | 76.46 | 90.00 | 74.17 | **90.00** | 95.20 | **97.35** | 97.86 | 99.20 | 63.71 | 83.33 | 61.76 | 75.27 | 78.20 | 89.19 |
| DoLa | 76.25 | 90.00 | 72.08 | 86.67 | 95.11 | **97.35** | 97.85 | 99.40 | 64.36 | **87.37** | 61.25 | 75.27 | 77.82 | 89.34 |
| ST | **80.00** | 90.00 | 72.29 | 83.33 | 95.25 | 96.59 | 97.65 | 98.60 | 64.90 | 77.78 | 59.41 | 71.33 | 78.25 | 86.27 |
| ST-G | 79.37 | 90.00 | 67.50 | **90.00** | 95.20 | 97.19 | 97.62 | 99.20 | 64.61 | 85.35 | 62.57 | 77.02 | 77.81 | 89.79 |
| Ours | 78.33 | 90.00 | **76.46** | **90.00** | 95.20 | **97.35** | 97.92 | 99.40 | 64.90 | 85.35 | **63.10** | 77.06 | **79.32** | **89.86** |
| ***MiMo-7B-RL*** | | | | | | | | | | | | | | |
| CoT | 65.21 | 83.33 | 59.17 | 80.00 | 94.16 | **97.19** | 95.35 | 98.80 | 51.36 | 86.36 | 56.01 | 71.68 | 70.21 | 86.23 |
| DoLa | **67.08** | 83.33 | 59.17 | 80.00 | 94.24 | 97.04 | 95.07 | 98.80 | 51.86 | **86.87** | 55.58 | **73.12** | 70.50 | 86.53 |
| ST | 56.67 | 83.33 | 54.79 | 80.00 | **94.70** | 96.36 | 94.06 | 98.40 | 51.99 | 80.30 | 54.39 | 72.76 | 67.77 | 85.19 |
| ST-G | 65.62 | 83.33 | 59.17 | 80.00 | 94.33 | 97.04 | **95.99** | 99.00 | 52.18 | 85.56 | 54.57 | 72.40 | 70.31 | 86.27 |
| Ours | 66.46 | 83.33 | **60.62** | **83.33** | 94.36 | 96.89 | 95.35 | 99.00 | 52.18 | 85.35 | **56.09** | 72.40 | **70.84** | **86.72** |
| ***Qwen3-30B-A3B-Thinking*** | | | | | | | | | | | | | | |
| CoT | 87.29 | 90.00 | 81.25 | 90.00 | 96.50 | **97.88** | 98.30 | 99.60 | 70.77 | 86.36 | 67.79 | 81.36 | 83.65 | 90.87 |
| DoLa | 87.50 | 90.00 | **82.08** | **93.33** | 96.50 | **97.88** | 98.31 | 99.60 | 71.53 | 87.88 | 68.35 | 79.21 | **84.04** | 91.32 |
| ST | 85.62 | 90.00 | 75.21 | 90.00 | **96.66** | 96.89 | 98.25 | 99.00 | 71.88 | 78.28 | 66.33 | 76.34 | 82.33 | 88.42 |
| ST-G | 86.46 | 90.00 | 79.79 | **93.33** | 96.50 | **97.88** | **98.35** | 99.60 | **72.06** | 87.88 | 68.03 | 82.08 | 83.53 | 91.79 |
| Ours | **87.92** | 90.00 | 81.04 | **93.33** | 96.48 | 97.73 | 98.30 | 99.60 | 71.09 | **91.94** | **69.11** | **82.80** | 83.92 | **92.48** |

*Table 2.* Generation length w.r.t different models and methods.

| | Qwen3-4B | MiMo | Qwen3-30B | Overall |
|---|---|---|---|---|
| CoT | 12,269 | 9,948 | 10,285 | 10,834 |
| DoLa | 12,214 | 9,956 | 10,171 | 10,780 |
| ST | 11,713 | 10,246 | 9,988 | 10,635 |
| ST-G | 11,993 | 10,024 | 10,226 | 10,748 |
| Ours | 12,277 | 10,064 | 10,428 | 10,923 |

mended hyperparameters. To assess generalizability, experiments are conducted on five models, spanning 4B to 32B parameters, covering different architectures (dense and Mixture-of-Experts (Shazeer et al., 2017)) and model families (Qwen (Yang et al., 2025), MiMo (Xiaomi et al., 2025), and Llama (Grattafiori et al., 2024)). For more implementation details, please refer to Appendix D.1.

### 4.2. Comparison to Baselines

To evaluate the effectiveness of our proposed method, we compare LED with strong baselines in Table 1 for pass@1 and pass@16, and in Table 2 for generation length. Across all models and benchmarks, LED achieves the best overall performance without significant extra generation length.

Our proposed method LED consistently outperforms all baselines, and in most cases, improves the CoT baseline. On average, it **improves pass@1 and pass@16 by 0.67 and 0.92 percentage points** over the benchmarks, while maintaining nearly identical generation length (increased

from 10,834 to 10,923, $< 1\%$). This indicates that LED restores LRMs' exploration capability without sacrificing pass@1 accuracy and efficiency.

We also evaluate LED on different temperatures, and the accuracy-temperature slope $\alpha$ with LED is illustrated in Figure 5. For all of the latest LRMs, applying LED effectively turns $\alpha$ from negative (CoT) to positive. The shift of $\alpha$ majorly comes from increased pass@$n$ on high temperatures and non-decreased pass@$n$ on low temperature (thanks to the exploitation branch). This finally demonstrates the core motivation of LED: **leveraging latent posteriors, the probability of informative next-token candidates could be effectively amplified for better exploration**.

For the baseline methods: *DoLa* improves pass@1 and pass@16 compared to the CoT baseline on most cases, highlighting the value of decoding with latent posteriors. *SoftThinking (ST)* achieves the shortest generation length. However, the almost deterministic decoding during the thinking phase results in significantly lower pass@16, indicating the collapse of exploration. *SoftThinking-Gumbel (ST-G)* introduces stochasticity via adding Gumbel-Softmax noise upon the final-layer posterior, and improves both pass@1 and pass@16 compared to SoftThinking, confirming the importance of exploration. However, these methods failed to consistently improves exploration capability, measured by pass@16, while maintaining pass@1 performance.

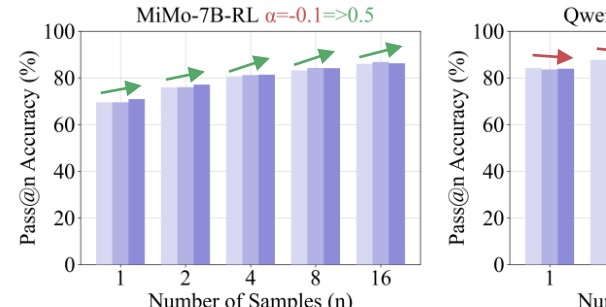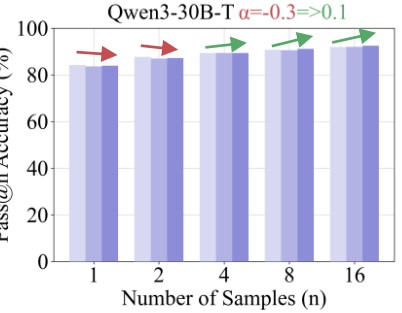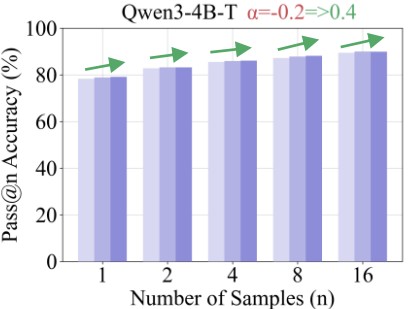

*Figure 5.* Pass@$n$ accuracy (%) for the latest LRMs with LED under varying sampling temperatures.

We also provide results on earlier LLMs DeepSeek-8B and QwQ-32B in Appendix D.2. LED is also effective to these models, but has relatively lower $\alpha$ gain.

*Table 3.* Ablation results on Qwen3-4B-Thinking.

| Setting | Pass@1 | Pass@16 | #Length |
|---|---|---|---|
| CoT | 78.20 | 89.19 | 12,269 |
| Ours | 79.32 | 89.86 | 12,277 |
| - w/o Think-only | 78.74 | 89.89 | 12,446 |
| - w/ LayerNorm | 77.97 | 90.65 | 12,510 |
| - w/ $p_{\text{top-}k}$ Norm | 77.92 | 88.45 | 13,070 |
| - w/o Exploitation | 64.63 | 83.24 | 16,350 |
| - w/o Top-$k$ Filtering | N/A | N/A | Maximum |

### 4.3. Ablation Study

To further investigate the effectiveness of the designs in LED, we conduct ablation studies to analyze their contribution. Main ablation results are summarized in Table 3.

**Exploration should be applied on *DeepThink* Only.** Removing the *DeepThink*-exploration-only strategy leads to a slight drop of pass@1 by 0.58 percentage point, almost unchanged pass@16 accuracy, and slightly increased generation length. This confirms that LED should only be applied on the *DeepThink* phase, as the following response generation requires less exploration.

**LayerNorm on latent hidden states encourages exploration, but worsens pass@1.** In practice (Wolf et al., 2020), the Final LayerNorm module is only applied on the final-layer hidden state $h^L$, not the latent ones (as DoLa's setting). However, the Final LayerNorm module could also be considered as a pre-processor of the LM-Head. Facing the dilemma, we conduct an experiment on both variants. Results show that applying the Final LayerNorm to the latent states degrades pass@1 by 1.35 percentage points, but provides 0.79 points performance gain on pass@16. The result could be supported by the observation in Figure 4: with the Final LayerNorm, the overall top-$k$ coverage ratios of latent posteriors are enhanced, bringing stronger valid signals, but on the other hand, introduces more instability. For fair comparison and reduced computation, we do not apply the Final LayerNorm to the latents in our default setting.

**Preserving the original $p_{\text{top-}k}^l$ before aggregation.** Similar to the Final LayerNorm, another variant of obtaining latent posteriors is: whether to normalize $p_{\text{top-}k}^l$ to *sum to one* before cumulative aggregation. An intuitive answer is no, as the absolute scale of the latent posteriors signifies how confident the predictions are, and manually normalizing them would amplify the confidence. Our results supports this: normalizing top-$k$ probabilities before cumulative aggregation worsens both pass@1 and pass@16 by 1.4 percentage points, and induces longer generation length.

**Balancing exploration with exploitation matters.** Removing the exploitation branch causes significant degradation, with pass@1 and pass@16 accuracy dropping sharply by 14.7 and 6.6 percentage points, respectively, and generation length increasing significantly by 33%. This highlights the importance of balancing exploration and exploitation.

**Top-$k$ filtering effectively guarantees generation sanity.** Removing final-layer top-$k$ filtering could cause generating nonessential tokens, thus lead to endless looping and most the generations reaching the context limit. We manually interrupted the experiments to reduce meaningless carbon emissions. These results shows that restricting exploration to calibrated candidates, i.e., the top-$k$ tokens of the top-layer posterior $p^L$, is critical to preserve generation sanity.

**LED is robust to exploration depth $d$.** Figure 7 analyzes the effect of exploration depth $d$ ($d = 1$ denotes regular decoding). From $n = 1$ to $n = 16$ express a similar trend: increasing $d$ first brings significant performance gain, and then tend to saturate at $d = 12$, where top-$k$ coverage ratio is close to zero, as illustrated in Figure 4). While very large $d$ could introduce noise and slightly reduce the performance. Across most settings, LED outperforms regular decoding (horizontal dashes), demonstrating LED's robustness to $d$.

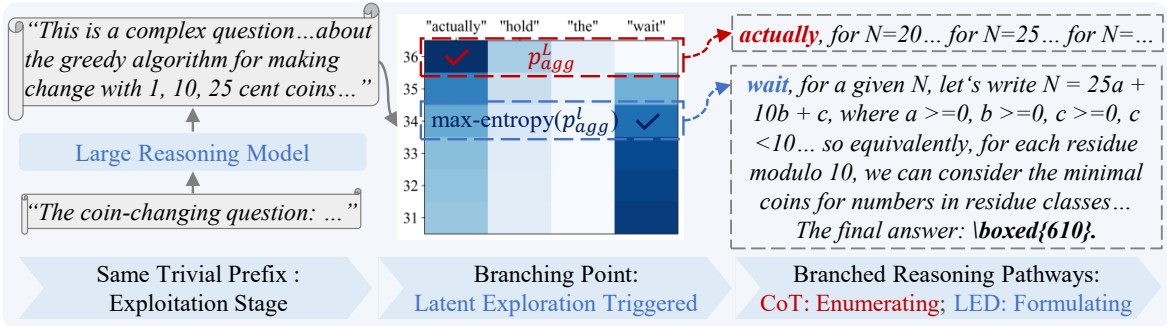

*Figure 6.* A case study of Qwen3-4B-Thinking with regular decoding (CoT) and our proposed LED on the AIME 2025 dataset.

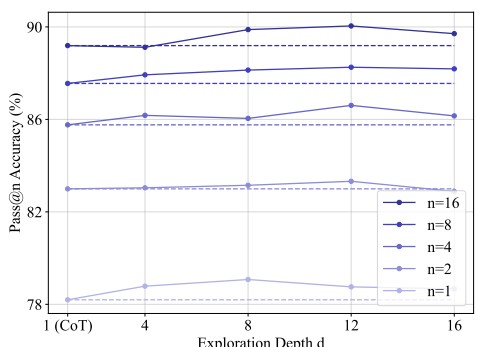

*Figure 7.* Pass@$n$ accuracy of of varying exploration depth $d$.

*Table 4.* Test accuracy on different training-test rollout strategies.

| Test \ Training Rollout | Untrained | Regular | LED |
|---|---|---|---|
| Regular | 31.25 | 41.99 | 43.10 |
| LED | 32.60 | 43.66 | **45.44** |

### 4.4. LED as a Rollout Policy for RL

LED's exploration motivation naturally extends to RL rollouts. To verify this, we apply LED during GRPO rollout, and compare it with regular decoding. Detailed experimental settings are deferred to Appendix D.3.

Table 4 shows that using LED during rollout consistently improves the final policy. Notably, models trained with LED outperform those trained with regular rollout. Even when evaluated with Regular decoding, the model trained with LED achieves higher accuracy (43.10 vs. 41.99), indicating that LED improves the learned policy itself rather than merely enhancing test-time search. These results suggest that LED is not limited to inference-time correction, but can also serve as an improved rollout policy for more effective exploration during RL training.

### 4.5. Case Study

To illustrate how LED leverages latent exploration, we conduct a zero-temperature case study on AIME 2025 using Qwen3-4B-Thinking. Compared with standard CoT, LED correctly solves one additional problem (Question #22), yielding a 3.3-point pass@1 improvement.

As shown in Figure 6, LED initially behaves identically to CoT during the early *DeepThink* stage, generating the same preliminary analysis. This indicates that LED remains exploiting when the final-layer posterior is highly confident.

The two methods diverge at an intermediate timestep. The final-layer posterior strongly favors the token "*actually*", which is selected by CoT. In contrast, intermediate latent posteriors retain non-trivial probability mass on "*wait*", indicating a potential reflection path. After aggregation, the highest-entropy latent posterior assigns slightly higher probability to "*wait*", which is therefore selected by LED.

This branching point leads LED to re-examine the problem and discover the key insight that greedy failure is determined by specific modulo-25 remainders, enabling an efficient solution. By contrast, CoT continues exhaustive enumeration and eventually exhausts the context window without reaching the correct answer.

## 5. Conclusion

In this paper, we first identify an entropy collapse phenomenon in final-layer posterior of modern RL-post-trained Large Reasoning Models. Through empirical analysis, we show that intermediate layers retain substantial entropy. Based on this insight, we propose Latent Exploration Decoding (LED), a simple and training-free method that restores exploration by aggregating latent posteriors from intermediate layers, and selecting the most informative depth via entropy. Extensive experiments demonstrate consistent improvements pass@1 to pass@16 accuracy across multiple models and benchmarks with negligible extra overhead.

*Acknowledgement* This work was supported by the Fundamental and Interdisciplinary Disciplines Breakthrough Plan of the Ministry of Education of China (No. JYB2025XDXM702).

## Impact Statement

This paper presents work whose goal is to advance the field of Machine Learning, particularly the understanding and decoding of large reasoning models. While enhanced reasoning performance may indirectly affect downstream applications, these impacts are consistent with other progress in language modeling systems. We do not foresee any unique or immediate societal risks arising specifically from this work.

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

# A. Related Work

## A.1. LLM Reasoning

Chain-of-Thought (CoT) prompting (Wei et al., 2022) enables large language models to perform explicit step-by-step reasoning, substantially improving performance on complex tasks such as mathematics, code generation, long-form writing, and multimodal understanding (Cao et al., 2025; Tan et al., 2025a; Li et al., 2025). Subsequent work introduced explicit reasoning phases, most notably *DeepThink* (Jaech et al., 2024; Guo et al., 2025), which requires models to generate internal reasoning traces within dedicated "$\langle$think$\rangle\langle$/think$\rangle$" tags prior to producing final answers.

Reinforcement learning (RL) based post-training further enhances reasoning performance by directly optimizing for correctness over sampled solutions. In particular, group-based RL methods such as Group Relative Policy Optimization (GRPO) (Shao et al., 2024) and its variants, including DAPO and GSPO (Yu et al., 2025; Zheng et al., 2025), train models by sampling multiple candidate answers, rewarding correct ones, and penalizing incorrect ones within each group. While highly effective at improving pass@1 accuracy, these methods implicitly favor confident and consistent outputs, which can reduce diversity in the surface-level output distribution. Our work complements this line of research by focusing on decoding rather than training, and by explicitly addressing the exploration collapse induced by RL post-training.

## A.2. LLM Decoding

Decoding methods for LLMs can be broadly categorized into two paradigms: (i) *hard decoding*, which samples discrete tokens from a modified posterior distribution, and (ii) *soft decoding*, which operates directly on continuous representations or embeddings.

**Hard decoding.**  Classical sampling strategies such as temperature scaling, top-$k$ sampling (Fan et al., 2018), nucleus (top-$p$) sampling (Holtzman et al., 2019), and min-$p$ sampling (Nguyen et al., 2024) operate directly on the final-layer posterior. These methods reshape or truncate the distribution while preserving the relative ordering of token probabilities. A related class of approaches explicitly modifies the posterior ordering. Contrastive Decoding (CD) (Li et al., 2023) contrasts the outputs of a large model with those of a smaller model to amplify factual knowledge present in the former. DoLa (Chuang et al., 2023) similarly exploits discrepancies across layers, leveraging the growth of factual knowledge within the network to adjust the final-layer posterior.

**Soft decoding.**  Soft decoding methods avoid discrete token sampling and instead propagate continuous representations across decoding steps. This paradigm is sometimes referred to as *continuous* or *latent* reasoning, particularly in the context of reasoning tasks. Coconut (Hao et al., 2024) proposes training a model that directly feeds the final-layer hidden state at timestep $t$ as the input embedding at timestep $t + 1$. CoLaR (Tan et al., 2025b) improves upon Coconut by introducing a reparameterization trick and applying GRPO-style RL training to stabilize latent reasoning. SoftThinking (Zhang et al., 2025) replaces hard token sampling with posterior-weighted embedding averaging, enabling a breadth-first-search-like exploration over the reasoning space. SoftThinking-Gumbel (Wu et al., 2025) further injects stochasticity via Gumbel-Softmax noise to enhance exploration diversity.

Our proposed Latent Exploration Decoding (LED) primarily falls within the *hard decoding* paradigm, while drawing inspiration from soft decoding approaches. LED differs from prior methods in two key aspects. First, LED targets exploration across *network depth*, rather than across the vocabulary alone. Second, unlike DoLa, which contrasts low-layers' posteriors to the final-layer posterior with maximum Jensen-Shannon Divergence (i.e., *most dissimilar*), to amplify factual knowledge, LED selects high-layers latent posterior combinations with maximal entropy (i.e., *most explorative*), to enable adaptive and effective exploration.

# B. Motivation

## B.1. Regular Decoding Process Explained

We briefly review the standard decoding process used in autoregressive language models, which serves as the baseline for our analysis and motivates the limitations addressed by LED.

Given an input prompt and a partially generated sequence, an LLM produces a sequence of hidden states $\{h^l\}_{l=1}^{L}$ through $L$ transformer layers. Decoding relies exclusively on the final-layer hidden state $h^L \in \mathbb{R}^d$ at the current time step. This hidden

state is projected to the vocabulary space via the language modeling head:

$$z^L = W h^L + b, \tag{7}$$

where $W \in \mathbb{R}^{|\mathcal{V}| \times d}$ and $b \in \mathbb{R}^{|\mathcal{V}|}$ denote the output projection parameters, and $\mathcal{V}$ is the vocabulary.

The resulting logits $z^L$ are converted into a probability distribution over next-token candidates by temperature-scaled softmax:

$$p^L(x \mid h^L, \tau) = \frac{\exp(z_x^L/\tau)}{\sum_{x' \in \mathcal{V}} \exp(z_{x'}^L/\tau)}, \tag{8}$$

where $\tau > 0$ denotes the sampling temperature. Lower temperatures sharpen the distribution toward its mode, while higher temperatures flatten the distribution and increase sampling diversity.

Optionally, additional truncation mechanisms such as top-$k$ or nucleus (top-$p$) filtering (Fan et al., 2018; Holtzman et al., 2019) may be applied to $p^L$. A next token $x_{t+1}$ is then sampled from the resulting distribution (or selected greedily when $\tau \to 0$), appended to the sequence, and fed back into the model to produce the next hidden state.

Crucially, standard decoding operates *solely* on the final-layer posterior $p^L$. All intermediate-layer hidden states $\{h^l\}_{l=1}^{L-1}$ are discarded after computing $h^L$, and thus any uncertainty or alternative hypotheses represented in earlier layers do not directly influence the sampling process. As a result, when RL post-training compresses the entropy of $p^L$, standard decoding lacks a mechanism to recover exploration, even if substantial latent uncertainty remains in intermediate representations.

### B.2. Mechanistic Explanation of Entropy Compression under Sparse Correctness Rewards

We provide a mechanistic explanation for why Group Relative Policy Optimization (GRPO) (Shao et al., 2024) post-training with sentence-level correctness rewards tends to reduce *token-level* entropy, with a disproportionate effect on high-variance branching tokens that are critical for exploration.

**Disclaimer.** This subsection provides theoretical intuition to motivate the empirical entropy measurements reported in the main text. It is not intended as a formal convergence proof; rather, it articulates the credit-assignment dynamics that drive empirical observations of entropy collapse in sparse-reward settings (Shao et al., 2024; Yu et al., 2025; Zheng et al., 2025).

**GRPO Objective and Sparse Rewards** Consider the GRPO objective (Shao et al., 2024), applied to a policy $\pi_\theta$ with a group size of $G$:

$$\mathcal{J}_{\text{GRPO}}(\theta) = \mathbb{E}_{\{o_i\}_{i=1}^G \sim \pi_{\theta_{\text{old}}}, q \sim \mathcal{Q}} \left[ \frac{1}{G} \sum_{i=1}^G \min \left( s_i A_i, \ \text{clip}(s_i, 1 - \epsilon, 1 + \epsilon) A_i \right) \right] - \beta D_{\text{KL}}(\pi_\theta \| \pi_{\text{ref}}), \tag{9}$$

where $s_i = \pi_\theta(o_i \mid q)/\pi_{\theta_{\text{old}}}(o_i \mid q)$ is the importance sampling ratio, and the advantage $A_i$ is computed as:

$$A_i = \frac{r_i - \text{mean}(\{r_j\}_{j=1}^G)}{\text{std}(\{r_j\}_{j=1}^G) + \mathbb{I}[\text{std} = 0]}, \tag{10}$$

with the convention that the denominator is set to 1 when all rewards in the group are identical (avoiding division by zero).

In mathematical reasoning tasks, the reward $r_i \in \{0, 1\}$ indicates whether the *entire generated sequence* $o_i$ is judged correct. Because the reward is sparse and applies only at the sequence level, the gradient signal concentrates on tokens whose variation most strongly correlates with the binary outcome.

**Asymmetric Credit Assignment at Branching Tokens** In autoregressive generation, not all positions contribute equally to sequence-level correctness. *Branching tokens*, typically early or structurally pivotal positions that select between qualitatively distinct solution paths, exhibit high variance in downstream outcomes. For example, in a mathematical proof, selecting an initial strategy (e.g., "Let $x$ denote..." vs. "Assume for contradiction..." vs. "By induction...") often determines whether the remainder of the solution can succeed at all.

Under the GRPO objective, the effective gradient on the policy at position $t$ is weighted by the group-relative advantage $A_i$. Because rewards are binary and sparse, $A_i$ is non-zero primarily for groups containing both correct ($r = 1$) and incorrect ($r = 0$) completions. Within such groups, the advantage differentiates sharply between tokens that lie on successful trajectories versus those on unsuccessful ones. Consequently:

1. Branching tokens that disproportionately determine correctness receive strong, consistent gradient pressure toward high-reward continuations.

2. Tokens downstream of a fixed solution path, whose variation affects only surface form rather than correctness, receive weaker or near-zero net advantage, yielding slower distributional sharpening.

This creates an *asymmetric credit assignment* mechanism: entropy reduction is rapid at high-variance decision points but gradual elsewhere.

**Token-Level Entropy Dynamics**   We define the conditional entropy at position $t$ given context $x_{<t}$ and query $q$ as:

$$H_t(\theta) = -\sum_{v \in \mathcal{V}} \pi_\theta(v \mid x_{<t}, q) \log \pi_\theta(v \mid x_{<t}, q).$$

Early in training, branching tokens typically exhibit high entropy due to the multiplicity of plausible exploration paths. As training progresses, the group-relative advantage amplifies the likelihood ratio $s_i$ for tokens that initiate high-reward branches. Because the sparse reward signal does not distinguish between stylistic variations *within* a successful branch, probability mass concentrates rapidly onto a small subset of high-value tokens at these critical positions, driving $H_t \to 0$.

In contrast, tokens that do not influence sequence-level correctness retain higher entropy for longer, resulting in a *differential* pattern of entropy reduction: sharp collapse at branching tokens, with comparatively mild sharpening at non-critical positions. This localized collapse is consistent with empirical observations of diversity reduction in sparse-reward RL and Reinforcement Learning from Human Feedback (RLHF) training (Ouyang et al., 2022).

**Limitations of KL Regularization**   The KL-divergence term $\beta D_{\mathrm{KL}}(\pi_\theta \,\|\, \pi_{\mathrm{ref}})$ anchors the optimized policy to the reference $\pi_{\mathrm{ref}}$, which typically exhibits higher entropy. In principle, for sufficiently large $\beta$, this term can prevent entropy collapse by constraining the policy to remain within a high-entropy neighborhood of the prior.

However, under the moderate $\beta$ values commonly employed in practice (Shao et al., 2024) (or even discarded (Yu et al., 2025)), where the KL penalty balances stability against reward optimization, the regularization primarily *slows* entropy reduction without preventing it. In this regime, the RL gradient (scaled by group-relative advantages) eventually dominates the KL gradient at high-impact tokens, causing $\pi_\theta$ to concentrate probability mass on empirically successful branches as training converges.

**Summary**   GRPO post-training with sparse, sentence-level correctness rewards induces *token-level* entropy compression through asymmetric credit assignment. Because the sparse reward signal differentiates most strongly at high-variance branching decisions, entropy collapses preferentially at these tokens, while other positions retain higher entropy for longer. KL regularization mitigates but typically does not eliminate this effect under standard training configurations. This mechanistic picture motivates the monitoring of token-level entropy dynamics during training and aligns with broader empirical findings on diversity collapse in language model RL.

### B.3. Experimental Setup for Statistics Analyzed

This section details how statistics are obtained and processed, for analyzing accuracy-temperature slope $\alpha$, Entropy-Layer trend, and top-$k$ coverage ratio $r_k$. All the corpus used for analyses are collected from the CoT generations across the benchmarks evaluated.

**Accuracy-temperature slope $\alpha$.**   We evaluate each model under three sampling temperatures $\{0.1, 0.3, 0.6\}$ and report the corresponding *pass@n* accuracies for $n \in \{1, 2, 4, 8, 16\}$. This yields a $3 \times 5$ accuracy matrix, which can be viewed as 15 points in a three-dimensional space defined by temperature, $n$, and accuracy.

To quantify how accuracy changes with temperature, we fit a **least-squares planar** approximation to this matrix. Specifically, we approximate accuracy as a linear function of temperature index and $n$ index. The resulting plane provides a smooth summary of accuracy trends across both dimensions. We define the accuracy-temperature slope $\alpha$ as the **slope of this plane along the temperature dimension**. A positive $\alpha$ indicates that higher temperatures improve accuracy, while a negative $\alpha$ reflects degraded performance under increased sampling randomness.

We also tested the models with higher temperatures $\{1.0, 2.0\}$ and found that, for earlier LLMs, setting temperature to 1.0 still enables consistent performance gain. However, for the LRMs, the performance deteriorate significantly. With temperature set to 2.0, all of the models fail to generate complete responses under most cases.

**Layerwise Analysis.** For each model, we randomly sample 30 questions from each dataset (noting that AIME 2024/2025 contains only 30 questions) and generate one response per question using standard Chain-of-Thought (CoT) decoding. This results in 120 samples spanning diverse domains, with approximately one million generated tokens per model in total.

At each generation step, we record all hidden states $\{h^l\}_{l=1}^L$ and pass them through the language modeling head to obtain the corresponding layerwise posteriors $\{p^l\}_{l=1}^L$. We compute the entropy of posteriors as

$$H(p^l) = -\sum_{i=1}^{V} p^l(i) \log p^l(i). \tag{11}$$

The final statistics reported in Figure 2 are averaged over all generation steps and questions.

# C. Methodology

## C.1. PyTorch Implementation.

To ensure fully reproducibility, we provide an executable PyTorch implementation of Latent Exploration Decoding, which is completely batchable and memory-efficient, thus could be executed under high concurrency:

**Core function of Latent Exploration Decoding**

```python
def latent_exploration_decoding(
    d_logits: torch.Tensor,
    topk: int,
    temperatures: torch.Tensor,
    is_thinking_mode: torch.Tensor,
    eps: float = 1e-6,
) -> torch.Tensor:
    B, d, V = d_logits.shape
    d_probs = torch.softmax(d_logits.div_(temperatures), dim=-1)
    origin_probs = d_probs[:, 0, :]  # (B, V), REVERSED for cumsum
    origin_topk_probs, origin_topk_ids = origin_probs.topk(k=topk, dim=-1)
    dk_probs = d_probs.gather(
        dim=-1, index=origin_topk_ids.unsqueeze(1).expand(-1, d, -1)
    ).clamp_min_(eps)  # (B, d, k)

    batch_next_origin_indices = torch.multinomial(
        origin_topk_probs, num_samples=1
    )  # (B, 1), exploit, automatically normalized
    origin_probs_top1 = origin_topk_probs[:, :1]  # (B, 1, k)
    do_exploration = torch.bernoulli(1 - origin_probs_top1).bool()  # (B, 1)
    accumulative_probs = torch.cumsum(dk_probs, dim=1)  # (B, d, k)
    accumulative_probs = accumulative_probs / accumulative_probs.sum(
        dim=-1, keepdim=True)  # (B, d, k)
    entropy = -torch.sum(accumulative_probs * torch.log(
        accumulative_probs.clamp_min_(1e-9)), dim=-1)  # (B, d)

    explore_layer = torch.argmax(
        entropy, dim=-1).view(B, 1, 1).repeat(1, 1, k)  # (B, 1, k)
    explore_probs = accumulative_probs.gather(
        dim=1, index=explore_layer).squeeze(1)  # (B, k)

    batch_next_topk_indices = torch.multinomial(
        explore_probs, num_samples=1).view(B, 1)
    batch_next_topk_indices = torch.where(
        do_exploration.logical_and(is_thinking_mode),
        batch_next_topk_indices,
        batch_next_origin_indices)
    batch_next_token_ids = origin_topk_ids.gather(
        dim=1,index=batch_next_topk_indices).squeeze(1)
    return batch_next_token_ids
```

Note that in practice, we **reverse** the hidden states and form $\{h^l\}_{l=L}^{L-d+1}$, thus it could be directly applied to "*torch.cumsum()*" without flipping along depth dimension (by calling "*torch.cumsum(dk_probs.flip(dims=(1,)), dim=1).flip(dims=(1,))*"), which could introduce extra memory cost.

# D. Experiment

## D.1. More Implementation Details

For better reproducibility, we illustrate all the implementation details.

**Infrastructure.** All experiments are conducted on a single Linux machine with eight NVIDIA H20 GPUs. We use SGLang (Zheng et al., 2024) backend (version 0.4.6) for efficient inference.

**Hyper-parameters.** We use the widely-used hyper-parameters for LRMs decoding (Yang et al., 2025): temperature=0.6, top-$p$=0.95, top-$k$=20 for all baseline methods. The random seed is set to 0 for all the experiments.

For LED, we set the exploration depth $d = 8$ for all the LRMs (even though $d = 12$ performs better on Qwen3-4B-Thinking, as illustrated in Section 4.3). This is mainly motivated by two reasons: (i) According to Figure 4, $d = 8$ is a *safe* depth that has not reached intermediate layers with very high entropy, and remains top-$k$ coverage ratio. (ii) As discussed in Section 3.5, the main bottleneck for computation and memory is feeding the hidden states to the LM-Head ($O(dV)$). A relative small $d$ alleviates both computation and memory overhead. Furthermore, setting $d = 8$ could be fully parallelized on modern GPU sets using Tensor-Parallelism (TP) technique.

Note that we do not apply top-$p$ in our default implementation, which could be time consuming in some cases. Instead, we approximate it by using a relatively smaller top-$k$=8. This brings no significant change to overall performance, but is slightly more efficient in computation and memory, and simpler in implementation. Relevant experimental results are shown in Table 6.

**Basline Methods.**

- DoLa (Chuang et al., 2023): We use the recommended setting of "DoLa-low", which collects every two layers of the hidden states from the lower 20 Transformer layers for contrastive decoding.

- SoftThinking (Zhang et al., 2025): Following SoftThinking's realeased paper and code, their performances are reproduced with early stopping entropy threshold and length set to 0.01 and 256, respectively, and soft-topk is set to 10.

- SoftThinking-Gumber (Wu et al., 2025): The setting of this method is nearly identical to SoftThinking's, with an extra Gumbel softmax noise temperature set to 0.5, as the official implementation recommended.

**Evaluation Benchmarks.**

- GSM8K (Cobbe et al., 2021) contains 1,319 grade-school level mathematics problems.

- MATH500 (Lightman et al., 2023) is a curated subset of 500 diverse problems from the MATH dataset (Hendrycks et al., 2021).

- The AIME2024 and AIME2025 benchmarks each include 30 problems from the American Invitational Mathematics Examination (AIME), providing challenging test cases that assess both accuracy and token efficiency.

- GPQA-Diamond (Rein et al., 2024) comprises 198 high-difficulty multi-choice questions across physics, chemistry, and biology domains.

- LiveCodeBench (Jain et al., 2024) is a dynamically updated code generation benchmark designed to prevent data contamination. Following SoftThinking, we use 279 problems published between August 2024 and January 2025.

**Prompts.** We use identical prompts to SoftThinking as follows:

> **Mathematical Reasoning**
>
> Please reason step by step, and put your final answer within \boxed{}.

*Table 5.* Pass@1 and pass@16 accuracy (%) across six benchmarks and two LRMs. We bold the **best** results and underline improved performance compared to the CoT baseline. ST, ST-G, GPQA-D, and LCB denote SoftThinking, SoftThinking-Gumbel, GPQA-Diamond, and LiveCodeBench, respectively.

| | Mathmatics | | | | | | | | Science | | Coding | | Overall | |
| | AIME2024 | | AIME2025 | | GSM8K | | MATH-500 | | GPQA-D | | LCB | | | |
|---|---|---|---|---|---|---|---|---|---|---|---|---|---|---|
| *DeepSeek-R1-Distill-Llama-8B* | | | | | | | | | | | | | | |
| CoT | 41.88 | **80.00** | 31.04 | 50.00 | 69.30 | 93.18 | 81.30 | 98.00 | 44.48 | **84.85** | **38.58** | 58.06 | 51.10 | 77.35 |
| DoLa | 43.13 | **80.00** | 31.04 | 60.00 | 69.65 | 93.71 | 81.16 | **98.40** | 44.63 | 82.83 | 38.13 | **60.57** | 51.29 | 79.25 |
| ST | 32.08 | 53.33 | 21.04 | 36.67 | **70.24** | 83.09 | 76.56 | 91.80 | 43.12 | 69.70 | 31.09 | 43.37 | 45.69 | 62.99 |
| ST-G | 43.54 | **80.00** | 30.21 | 63.33 | 68.85 | 93.86 | **82.03** | 98.00 | 44.63 | 81.31 | 37.77 | 58.78 | 51.00 | 79.21 |
| Ours | **45.52** | 76.67 | **33.54** | **70.00** | 68.55 | **94.84** | 79.76 | 97.80 | 44.16 | 81.82 | 38.49 | 58.78 | **51.65** | **79.99** |
| *QwQ-32B* | | | | | | | | | | | | | | |
| CoT | 78.12 | **90.00** | 68.33 | **90.00** | 95.70 | 97.35 | **97.21** | 99.20 | **64.68** | 89.39 | 61.85 | 74.91 | 77.65 | **90.14** |
| DoLa | 71.46 | 83.33 | 60.42 | **90.00** | 94.91 | 97.50 | 95.60 | 99.20 | 59.82 | **90.40** | 53.52 | 70.97 | 72.62 | 88.57 |
| ST | 75.21 | **90.00** | 64.58 | 83.33 | **95.84** | 96.66 | 96.94 | 98.80 | 62.03 | 81.82 | 57.91 | 70.97 | 75.42 | 86.93 |
| ST-G | 77.08 | **90.00** | 68.33 | 86.67 | 95.69 | 97.73 | 97.02 | **99.40** | 62.97 | 87.37 | 60.66 | **75.27** | 76.96 | 89.41 |
| Ours | **78.96** | **90.00** | **71.67** | **90.00** | 95.70 | **97.80** | **97.21** | 99.00 | 63.42 | 87.88 | **61.90** | 74.55 | **78.14** | 89.87 |

---

**Multi-Choice Questions (GPQA-Diamond)**

Please solve the following multiple-choice question. Please show your choice in the answer field with only the choice letter, e.g., "answer": "C".

---

**LiveCodeBench**

You will be given a question (problem specification) and will generate a correct Python program that matches the specification and passes all tests.
Read the inputs from stdin solve the problem and write the answer to stdout (do not directly test on the sample inputs). Enclose your code within delimiters as follows. Ensure that when the python program runs, it reads the inputs, runs the algorithm and writes output to STDOUT.

---

**Answer Verifier.** For coding questions, generated code is extracted by regex rules and verified in LiveCodeBench's official environment. For mathematical and multi-choice questions, we use HuggingFace's *math-verify* package[1] (version 0.7.0) to extract and compare final answers from long-form model responses using a set of predefined parsing rules. *The system is designed to minimize false positives/negatives, but they may occasionally occur.*

### D.2. Experimental Results on QwQ-32B and DeepSeek-8B

We also evaluate LED against the baseline methods on earlier LRMs, QwQ-32B and DeepSeek-8B (DeepSeek-R1-Distill-Llama-8B), as mentioned in Figure 1 and Section 4.2. The results are illustrated in Table 5. LED improves both pass@1 and pass@16 accuracy on DeepSeek-8B for 0.55 and 2.64 percentage points, respectively, and improves pass@1 accuracy on QwQ-32B for 0.49 percentage points. However, none of the decoding methods (DoLa, ST, ST-G, and our proposed LED) improves the overall pass@16 accuracy on QwQ-32B.

We explain the failure of decoding methods on QwQ-32B with one figure and two potential reasons: as illustrated in Figure 8, the Entropy-Layer curve of QwQ-32B express different tendency than other LRMs. (i) As one of the earliest released LRMs, QwQ-32B has not already been heavily post-trained (than Qwen3 series and MiMo-RL), thus their final-layer entropy has not collapsed. Especially, the entropy increases on the last layer, which could lead to the failure of LED. (ii) The overall Entropy-Layer curve is highly different from other LRMs we evaluated: the entropy of QwQ-32B decrease in early layers sharply, and then increases a bit and fluctuates, finally converges until the second to the last layer. The instability could cause DoLa's failure, which is designed under the hypothesis that factual knowledge "grows" among LLM layers.

---

[1] https://github.com/huggingface/Math-Verify

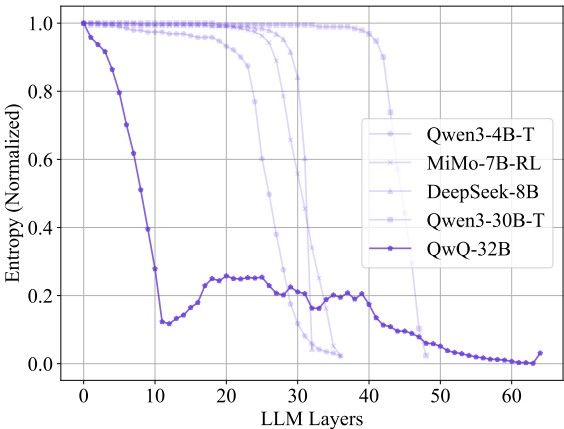

*Figure 8.* Normalized entropy across LLM layers.

### D.3. LED as a Rollout Policy for RL

Beyond test-time decoding, we further investigate whether LED can improve reinforcement learning (RL) training by serving as a stronger rollout policy.

**Experimental setup.** We initialize from Qwen3-4B-Thinking and conduct GRPO training implemented with VeRL and SGLang on 8×H100 GPUs. Following the DAPO setting, we apply all-easy-question filtering on `DigitalLearningGmbH/MATH-lighteval`, resulting in approximately 750 training samples and 750 test samples. We compare two rollout strategies during training: (1) standard Regular decoding, and (2) LED-based decoding. All other training hyperparameters are kept unchanged.

**Main results.** Table 4 reports the final test accuracy under different training and evaluation strategies. Models trained with LED rollouts consistently outperform those trained with Regular rollouts under both evaluation methods. Notably, even when evaluated with Regular decoding, the model trained with LED achieves higher accuracy (43.10 vs. 41.99), indicating that LED improves the learned policy itself rather than merely improving test-time search. The best performance is obtained when LED is used in both training and evaluation, reaching 45.44.

**Training dynamics.** Figure 9 shows that LED rollout leads to faster reward growth during GRPO training, suggesting more effective exploration and higher-quality trajectories. Figure 10 further shows that LED consistently yields better validation accuracy before and after RL training.

Interestingly, LED also slightly improves training efficiency in our setup. Training with Regular rollout takes approximately 4.87 hours, while LED rollout reduces total training time to 4.44 hours. This is mainly because LED generates shorter responses on average during training (approximately 12.7k → 12.1k tokens), reducing rollout cost despite the modest additional decoding overhead.

Overall, these results suggest that LED is not limited to post-hoc inference-time correction. Instead, it can also serve as an improved rollout policy that encourages more effective exploration during RL training.

### D.4. Experimental Results on Different Top-$k$ and Top-$p$

We compare CoT and LED under different top-$k$ and top-$p$ values, and the results are shown in Table 6. The results demonstrates that setting a relatively smaller top-$k$ value with discarded top-$p$ threshold expresses basically comparable performances on CoT and LED. Specifically, with a smaller top-$k$ value, both models expresses relatively higher (yet not significant) pass@1 accuracy and lower pass@16 accuracy. For computational efficiency, we discard the top-$p$ restriction in the default LED setup.

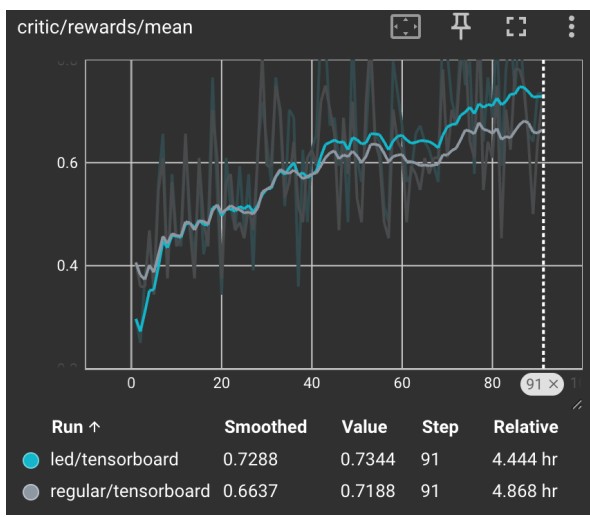
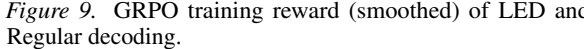
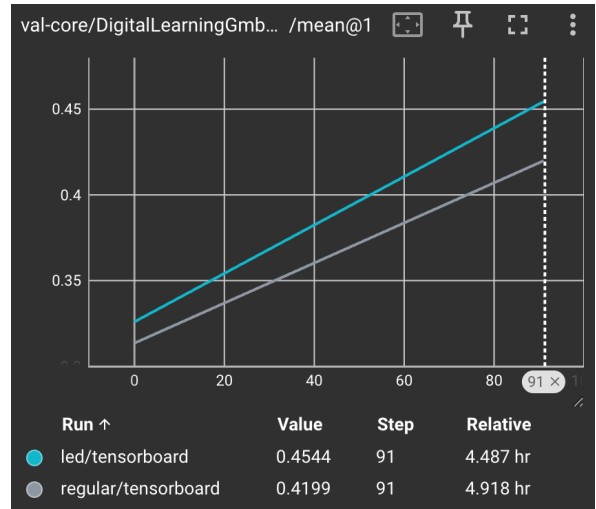

*Figure 9.* GRPO training reward (smoothed) of LED and Regular decoding.

*Figure 10.* Validation accuracy of LED and Regular decoding before and after GRPO training.

*Table 6.* Average pass@1, pass@16, and generation length of CoT and LED across benchmarks on Qwen3-4B-Thinking.

|  | CoT | | | LED | | |
|---|---|---|---|---|---|---|
| Top-$k$=20, Top-$p$=0.95 | 78.20 | 89.19 | 12269 | 79.30 | 89.88 | 12272 |
| Top-$k$=8, Top-$p$=1.00 | 78.24 | 89.12 | 12265 | 79.32 | 89.86 | 12277 |

# E. Efficiency Study

In this section, we provide additional efficiency analysis of LED, including wall-clock latency breakdown and high-concurrency throughput benchmarks.

## E.1. Wall-Clock Latency

We benchmark Regular decoding, DoLa, SoftThinking, and LED under batch size 4 on 8×A100 GPUs.

Table 7 reports the per-token latency breakdown over embedding, backbone forward, LM-Head, and sampling stages under 8K and 16K contexts.

LED incurs only modest runtime overhead compared with Regular decoding, while remaining consistently more efficient than both DoLa and SoftThinking. The additional overhead mainly comes from applying the LM head to intermediate hidden states, which modestly increases the LM-Head stage.

## E.2. High-Concurrency Throughput

We further benchmark decoding throughput under high concurrency: batch size 128, 16K context length, on 8×H100 GPUs.

LED achieves 91.81% of Regular decoding throughput while substantially outperforming DoLa and SoftThinking in efficiency. Combined with the accuracy gains reported in the main text, these results suggest that LED offers a favorable accuracy-efficiency tradeoff among exploration-oriented decoding methods.

*Table 7.* Wall-clock latency breakdown (ms/token).

| Strategy | Context | Embedding | Backbone | LM-Head | Sampling |
|---|---|---|---|---|---|
| Regular | 8K | 0.2025 | 5.5591 | 0.0807 | 0.1842 |
| DoLa | 8K | 0.1981 | 5.4885 | 0.1494 | 0.3771 |
| SoftThinking | 8K | 0.2704 | 5.5901 | 0.0857 | 0.5428 |
| LED | 8K | 0.2024 | 5.5332 | 0.1057 | 0.2361 |
| Regular | 16K | 0.1900 | 5.5674 | 0.0796 | 0.1796 |
| DoLa | 16K | 0.1940 | 5.6129 | 0.1434 | 0.3516 |
| SoftThinking | 16K | 0.2593 | 5.6844 | 0.0919 | 0.4877 |
| LED | 16K | 0.2049 | 5.6981 | 0.1085 | 0.2363 |

*Table 8.* High-concurrency throughput benchmark.

| Strategy | Regular | DoLa | SoftThinking | LED |
|---|---|---|---|---|
| Throughput (tok/s) | 4578 | 3362 | 3025 | 4204 |
| Relative (%) | 100.0 | 73.4 | 66.1 | 91.8 |

