# OpenReview forum: "Restoring Exploration after Post-Training: Latent Exploration Decoding for Large Reasoning Models"
_ICML.cc/2026/Conference — ICML 2026 regular_

### Official Review · Reviewer_4M26 · 2026-03-07

**Soundness:** 2
**Presentation:** 3
**Significance:** 2
**Originality:** 3
**Overall Recommendation:** 5
**Confidence:** 4

**Summary:**

This paper identifies an "exploration collapse" phenomenon in Large Reasoning Models (LRMs) that have undergone Reinforcement Learning post-training, where traditional temperature-based sampling fails to improve pass@n accuracy. The authors observe that while the final-layer posterior exhibits sharply reduced entropy, intermediate layers—termed "latent entropy reservoirs"—retain substantial uncertainty. To address this, they propose Latent Exploration Decoding (LED), a training-free strategy that aggregates intermediate posteriors and selects the depth configuration with maximal entropy for exploration. Applied exclusively during the "DeepThink" phase, LED consistently improves pass@1 and pass@16 across multiple reasoning benchmarks and models with negligible computational overhead.

**Compliance With Llm Reviewing Policy:**

Affirmed.

**Final Justification:**

i vote for accepting the paper

**Key Questions For Authors:**

I would be willing to consider increasing my score if the authors can provide rebuttal evidence or new experimental results demonstrating that incorporating LED into the RL rollout/training process leads to a more robust final policy compared to standard GRPO-style training. Proving that this method can enhance learning and not just decoding would significantly elevate the paper's contribution.

**Limitations:**

see above

**Strengths And Weaknesses:**

# strength
The paper provides a compelling mechanistic explanation and empirical evidence for why RL post-training (like GRPO) causes entropy collapse at the final layer while sparing intermediate layers. LED is a plug-and-play, training-free method that requires no additional parameters or model fine-tuning.The method demonstrates consistent improvements across diverse model families (Qwen, MiMo, Llama) and architectures (Dense and MoE), successfully "reactivating" the effectiveness of high-temperature sampling.

# weakness
The primary problems of this work is that the proposed method currently functions as a "post-hoc patch" rather than a fundamental solution to the limitations of RL post-training. By focusing solely on test-time decoding, the paper treats the entropy collapse of the final layer as an inevitable outcome to be mitigated, rather than a training deficiency to be corrected.

The effectiveness of LED is inherently tied to the fact that the model’s final layer is "over-optimized" or "squeezed". If future RL algorithms evolve to better preserve output diversity or incorporate more effective entropy regularization, the utility of a latent-layer decoding fix may diminish significantly.

**A significant missed opportunity in this research is the lack of experiments exploring the integration of LED into the RL rollout phase.** Currently, RL models are trained using rollouts that suffer from the very "exploration collapse" the authors describe. If the LED strategy were used during training to sample more diverse and potentially correct reasoning pathways, it could provide the RL algorithm with higher-quality gradients and a broader search space, potentially resulting in a superior base policy. The absence of such experiments makes it unclear whether this method can improve how models learn, or if it is strictly a tool for better "searching" a flawed model at inference time.

While the maximum entropy criterion is an intuitive heuristic for exploration, it lacks a rigorous theoretical guarantee that the highest-entropy layer contains semantically valid reasoning. In some layers, high entropy might simply represent unformed features or noise rather than a viable reasoning path.

---

> ### Author Rebuttal · Authors · 2026-03-31
>
> Dear Reviewer #4M26,
>
> Thank you for the thoughtful feedback and especially for the concrete suggestion on integrating LED into RL rollout/training. We also acknowledge that this is an important direction, and we therefore conducted a new experiment to test exactly this question.
>
> ---
>
> > New experiment: LED in RL rollout / training
>
> We conducted a GRPO-style training experiment in which LED is used during rollout generation instead of regular decoding. **[Experimental plots could be found in this anonymous webpage](https://anonymous.4open.science/r/LED-Rebuttal/README.md)**.
>
> Setup: starting from Qwen3-4B-Thinking, we run GRPO-style training with either Regular decoding or LED during rollout generation, using VeRL + SGLang on 8$\times$H100 GPUs. We apply DAPO-like all-easy filtering on DigitalLearningGmbH/MATH-lighteval, yielding about 750 train and 750 test samples.
>
> Final test accuracy (also partially shown in Figure 1.3):
>
> | Test \ Training Acc. | Untrained | Regular rollout | LED rollout |
> |---|---:|---:|---:|
> | Regular decoding | 31.35 | 41.99 | 43.10 |
> | LED decoding | 32.60 | 43.66 | **45.44** |
>
> These results show that LED improves not only inference-time search, but also the learned policy itself:
> 1. **Better final policy**: training with LED rollout improves the final model over standard rollout training, regardless of whether evaluation uses Regular or LED decoding.
> 2. **Faster learning** (Figure 1.1 and Figure 1.2): the reward/accuracy curve rises faster with LED rollout.
> 3. **Higher efficiency** (Figure 1.4): LED rollout actually reduces training time in this setup (4.87h $\rightarrow$ 4.44h), because it generates shorter responses on average (about 12k $\rightarrow$ 11k tokens).
>
> Therefore, LED is not only a post-hoc search patch for a fixed model; it can also serve as a stronger rollout policy that improves RL learning.
>
> > W2 / Q1: if future RL preserves diversity better, LED may lose value
>
> We agree that LED is especially motivated by the strong final-layer concentration observed in current post-trained reasoning LRMs. Importantly, **this is not an isolated artifact of one model**: in the manuscript, we consistently observe the same layerwise pattern across multiple modern reasoning models, namely that the final layer becomes much more concentrated while intermediate layers still preserve substantial uncertainty. Therefore, our main manuscript contribution is already to identify and exploit this widespread post-training phenomenon at decoding time.
>
> **The new RL-with-LED experiment should be viewed as a strengthening extension**: since this collapse pattern is common in current RL-style post-training, it is natural to ask whether the same latent-exploration mechanism can also improve rollout generation during RL itself, and our new results suggest that it can.
>
> > W3: maximum entropy may select noisy layers rather than semantically valid reasoning
>
> We agree that maximum entropy alone is not a correctness guarantee. LED is designed to avoid exactly the failure mode you pointed out: **it does not select an arbitrary high-entropy layer over the full vocabulary**. Instead, it first restricts exploration to the final-layer top-$k$ candidate set, so the latent uncertainty is always grounded in tokens already considered plausible by the final layer. The aggregation step then combines this final-layer semantic constraint with the residual uncertainty in intermediate layers, which serves as a practical denoising mechanism. Under this view, entropy is used only as a pragmatic criterion for selecting the least-collapsed point among **semantically constrained candidates**.
>
> ---
>
> We sincerely thank you again for the suggestion. We believe this new RL-rollout result substantially strengthens the paper by showing that LED can improve learning as well as decoding, and we will include these results and the corresponding discussion in the revision. **We will also open-source the RL training code when this paper is accepted**.

---

> > ### Author Rebuttal · Reviewer_4M26 · 2026-04-01
> >
> > I am very pleased with the new experiment of RL. I hope the author can keep their open-source promises. I also recommend other reviewers to raise the score. Good luck

---

> > > ### Author Response · Authors · 2026-04-02
> > >
> > > Thank you very much for the encouraging follow-up and for raising your score! We are especially glad that the new RL experiment addressed your main concern. We also appreciate your recommendation to other reviewers.
> > >
> > > We will fully open-source our code, including manusciript and rebuttal experiments, so that everyone could reproduce our results and use it for further research, after this paper is accepted.
> > >
> > > Authors of paper #3484

---

### Official Review · Reviewer_TaAK · 2026-03-11

**Soundness:** 2
**Presentation:** 3
**Significance:** 2
**Originality:** 3
**Overall Recommendation:** 2
**Confidence:** 3

**Summary:**

The paper investigates the exploration collapse phenomenon in Large Reasoning Models (LRMs) that have undergone Reinforcement Learning (RL) post-training. The authors observe that for these models, increasing sampling temperature no longer reliably improves pass@n accuracy because the final-layer posterior becomes overly confident and low-entropy. However, they discover that intermediate layers preserve higher entropy. To exploit this, the authors propose Latent Exploration Decoding (LED), a training-free decoding strategy. LED computes latent posteriors from intermediate layers via early exiting, filters them using the final-layer top-k candidates, aggregates them using a cumulative sum, and selects the distribution with the maximum entropy to sample from during the exploration phase. Experiments demonstrate improvements in pass@1 and pass@16 accuracy across several reasoning benchmarks

**Compliance With Llm Reviewing Policy:**

Affirmed.

**Key Questions For Authors:**

1. Can you provide a detailed analysis of the decoding latency and FLOPs overhead of LED compared to standard decoding? How does applying the LM-Head to intermediate layers  impact memory bandwidth during autoregressive generation?

2. What is the theoretical justification for using a final-to-latent cumulative sum for aggregation? Why would summing probability masses across different representational depths yield a meaningful exploration distribution?

**Limitations:**

No, the authors have not adequately discussed the limitations.

**Strengths And Weaknesses:**

### Strengths

1. The identification of exploration collapse in modern RL-trained LRMs (where temperature scaling fails) and linking it to the entropy asymmetry between intermediate and final layers is an interesting observation.

2. The empirical motivation effectively sets up the problem, showing how the final-layer posterior becomes squeezed due to RL objectives.

3. The paper is generally well-structured.

### Weaknesses

1. Computational Overhead. The claim of negligible inference overhead in the paper is unsupported. To compute the latent posteriors, the model must project intermediate hidden states through the language modeling head. In modern LLMs with large vocabularies, the LM-Head projection is an extremely memory-bandwidth and compute-intensive matrix multiplication. Doing this for multiple layers at every decoding step will substantially degrade generation tokens-per-second. The authors only report generation length, without wall-clock time or FLOPs.

2. Heuristic Design. The core mechanism of LED, applying a cumulative sum aggregation across layers and then selecting the one with maximum entropy, is highly heuristic. There is no rigorous theoretical justification for why a cumulative sum of probabilities across varying depth levels forms a principled probability distribution for exploration.

3. The empirical gains are marginal, increasing pass@1 and pass@16 accuracy by 0.61 and 1.03 percentage points. Given the likely substantial decoding latency overhead introduced by multiple LM-Head passes, it is unclear whether LED is practically useful compared to simply spending that compute on generating an additional standard sample.

4. The paper misses comparisons against more robust sampling techniques designed to handle entropy collapse, such as dynamic temperature scaling, or simply applying an entropy-penalty during decoding.

---

> ### Author Rebuttal · Authors · 2026-03-31
>
> Dear Reviewer #TaAK,
>
> Thank you for the detailed feedback. We agree that the original submission did not provide sufficient evidence on runtime overhead, and that the wording around the aggregation rule should be made more precise. We address these points below. **[All figures are provided here](https://anonymous.4open.science/r/LED-Rebuttal/README.md)**.
>
> ---
>
> > W1&Q1: computational overhead is unsupported
>
> We agree. Reporting only generation length is insufficient, so we additionally benchmarked wall-clock latency and high-concurrency throughput：
> ### Wall-Clock Benchmarking (batch size=4, 8*A100 GPUs, 16K context)
> | Strategy | Embed (ms/tok) | Backbone (ms/tok) | LM-Head (ms/tok) | Sampling (ms/tok) |
> |-|-:|---:|---:|---:|
> |Regular | 0.2025 | 5.5674 | 0.0796 | 0.1796 |
> |DoLa | 0.1940 | 5.6129 | 0.1434 | 0.3516 |
> |ST | 0.2593 | 5.6844 | 0.0919 | 0.4877 |
> |**LED**|**0.2049**|**5.6981**|**0.1085**|**0.2363**|
>
> ### High-Concurrency Throughput (batch size=128, 8*H100 GPUs, 16K context)
> |Strategy|Regular|DoLa|ST|**LED**|
> |-|-:|-:|-:|-:|
> |Througput (tok/s)|4578|3362|3025|**4204**|
> |Rel. (%)|100|73.4|66.1|**91.8**|
>
> We will switch to a more accurate claim: **LED has modest runtime overhead but is more efficient than the strong yet complex decoding baselines**.
> > W2 & Q2: the cumulative-sum + max-entropy mechanism is heuristic
>
> We agree that this component is not derived from a formal optimality theorem, and we will revise the wording.
>
> Our intent is not to claim that the cumulative sum defines one exact Bayesian posterior. Rather, it is an empirically motivated construction under two constraints observed in post-trained LRMs: shallow layers are often too immature, while the final layer is already over-collapsed.
>
> The starting point was the latent-reservoir intuition, where a weighted average would be the most immediate alternative. However, introducing **layer-wise weights in a training-free method would add extra hyperparameters and risk test-set overfitting, so we instead sought a hyperparameter-free rule**. Motivated by the fact that **deeper posteriors are generally more semantically aligned with the trained model behavior**, we aggregate from the final layer backward, so that later **more reliable layers are naturally retained more strongly in the final candidate distribution**.
>
> Maximum entropy is then used only as a pragmatic criterion for selecting the least-collapsed point among these semantically aggregated candidates.
>
> > W3: gains are marginal
>
> We agree that the absolute gains are modest, and we do not intend to overclaim LED as a large-margin improvement. Our main claim is about **consistent overall improvements** in a **strong-baseline, low-headroom regime**.
>
> To verify this rigorously, we ran 16 independent decoding trials per model and applied a one-sided Wilcoxon signed-rank test. LED's overall improvement over Regular decoding is **statistically significant** across all tested models ($p < 0.01$, one-sided, $n=16$), confirming that the gains are **reliable and not attributable to sampling variance**.
>
> After adding runtime measurements at W1, LED is substantially more efficient than DoLa and SoftThinking, while remaining close to regular decoding throughput. In this setting, the practical question is not only whether gains are small in absolute terms, but whether the method provides a favorable gain-per-cost tradeoff. Our new results support that it does.
> > W4: missing comparisons to dynamic temperature scaling
>
> Thank you for this suggestion. We additionally reproduced Dynamic Temperature Sampling (EDT)  [1] using the settings recommended in their paper: $T_0=1$, $N=0.8$, and $\theta=0.1$. The results are shown below (Q and M denote Qwen3 and MiMo, respectively):
>
> ||Q4B-p@1|M7B-p@1|Q30B-p@1|Q4B-p@16|M7B-p@16|Q30B-p@16|
> |-|-|-|-|-|-|-|
> |Regular|78.20|70.21|83.65|89.19|86.23|90.87|
> |EDT|77.91|68.59|83.69|89.25|85.04|91.33|
> |LED|**79.32**|**70.84**|**83.92**|**89.86**|**86.72**|**92.48**|
>
> EDT sometimes improves pass@16 but is unstable on pass@1, because **EDT treats all candidates equally thus can amplify noise**, whereas LED restores exploration from intermediate latent posteriors that still preserve useful uncertainty, and thus **LED amplifies effective signal** rather than merely flattening the final distribution. We will include this comparison in the revision.
>
> ---
>
> Thank you again for you insightful comments! We will add a dedicated limitations section stating: the absolute accuracy gains are modest though statistically significant; the cumulative-sum aggregation and max-entropy depth selection are empirically motivated heuristics; and earlier models like QwQ-32B benefit less. We will also incorporate the throughput results, EDT comparison, and a more precise statement of efficiency.
>
> [1] Zhang et al., EDT: Improving Large Language Models' Generation by Entropy-based Dynamic Temperature Sampling, arXiv 2024.

---

> > ### Author Rebuttal · Reviewer_TaAK · 2026-04-04
> >
> > Thank  the authors for the rebuttal, which addressed some of the concerns. But I have concerns remain for the following:
> >
> > 1. Unprincipled heuristic design: the core mechanism of the paper, applying a cumulative sum of probabilities across varying depth levels and selecting the one with maximum entropy, lacks theoretical justification. This makes the paper unprincipled and weakens the contribution of the method.
> >
> > 2. Unfavorable cost-to-benefit tradeoff: I appreciate the new high-concurrency throughput numbers. However, they reveal an 8.2% reduction in throughput (from 4578 to 4204 tok/s). Incurring an ~8% penalty in decoding efficiency for marginal absolute accuracy gains (0.61% to 1.03%) does not present a compelling practical advantage.
> >
> > Because of the unprincipled heuristic and very marginal empirical gains by the computation cost, I would like to keep the score.

---

> > > ### Author Response · Authors · 2026-04-05
> > >
> > > Thank you again for the careful follow-up and for engaging seriously with our rebuttal. We understand your remaining concerns regarding the heuristic nature of the aggregation rule and the cost-to-benefit tradeoff.
> > >
> > > We would still emphasize that LED is motivated by repeated cross-domain and cross-model observations, and that it achieves a favorable tradeoff relative to strong decoding baselines such as DoLa, SoftThinking (ST), and ST-G in both accuracy and efficiency. Nevertheless, we appreciate your perspective and the care with which you evaluated the work.
> > >
> > > Authors of paper #3484

---

### Official Review · Reviewer_PVJC · 2026-03-13

**Soundness:** 2
**Presentation:** 3
**Significance:** 2
**Originality:** 3
**Overall Recommendation:** 4
**Confidence:** 4

**Summary:**

This paper studies exploration collapse in RL-post-trained large reasoning models (LRMs). The authors argue that post-training sharpens the final-layer posterior such that increasing sampling temperature no longer improves pass@n, while intermediate-layer posteriors retain higher entropy. Based on this observation, the paper proposes Latent Exploration Decoding, a training-free decoding method that aggregates intermediate-layer posteriors, selects the highest-entropy aggregate, and switches between exploration and exploitation based on final-layer confidence. Across multiple reasoning benchmarks and models, the paper reports performance for pass@1 and pass@16 with little generation-length overhead.

**Compliance With Llm Reviewing Policy:**

Affirmed.

**Final Justification:**

Most of my concerns are addressed, i have slightly raised my score.

**Key Questions For Authors:**

1. How is the hyperparameter d chosen per model in the main experiments? Is it tuned on a validation set, fixed by a heuristic, or chosen manually?
2. In Table 3, “w/ LayerNorm” improves pass@16 but hurts pass@1, and “w/o Exploitation” hurts a lot. Could the authors give a more concrete explanation of when these variants help or fail? For example, do they mostly affect early reasoning tokens, branch points, or long-horizon reasoning traces?
3. Why is maximum entropy the best depth-selection criterion versus other plausible alternatives?
4. Can the authors report wall-clock latency or throughput overhead under some basic inference setup?

**Limitations:**

The paper should more explicitly acknowledge:
- the improvement gains are small (roughly +0.6 pass@1 and +1.0 pass@16 on average)
- many Table 1 entries are flat, negligible, or negative
- potential sensitivity to decoding hyperparameters and baseline tuning
- LED also appears to help earlier models, which weakens the framing of “restoration after post training”.

**Strengths And Weaknesses:**

**Strengths:**

1. Clear problem framing.
2. Simple, training-free method with intuitive motivation.
3. Broad evaluation across several benchmarks and models.
4. Solid ablations for major design choices.
5. Consistent gains with minimal extra generation length.

**Weaknesses:**

1. The empirical gains are very small. The paper itself summarizes the average improvement as only about +0.61 to +0.67 pass@1 and +0.92 to +1.03 pass@16 overall. More specifically, the improvements in Table 1 are often 0.1-0.7 points, zero, or negative. For example, on Qwen3-30B, overall pass@1 improves only from 83.65 to 83.92 (+0.27), and on several benchmark/model pairs the difference is effectively negligible (e.g., 95.20 -> 95.20, 97.86 -> 97.92, 98.30 -> 98.30, 96.50 -> 96.48). In addition, a non-negligable fraction of entries are worse than CoT or DoLa. In particular, LED is outperformed by DoLa in some overall metrics (e.g., Qwen3-30B overall pass@1: 83.92 vs 84.04), and several benchmark-level cells are flat or worse.
2. The causal claim about RL-induced exploration collapse is suggestive but not fully established.
3. Computational overhead claims would be better supported by wall-clock latency/throughput measurements.

---

> ### Author Rebuttal · Authors · 2026-03-31
>
> Dear Reviewer #PVJC,
>
> Thank you for the concrete concerns. We address each point below.  **[All figures are provided here](https://anonymous.4open.science/r/LED-Rebuttal/README.md)**.
>
> ---
>
> > W1: small empirical gain
>
> We agree that the absolute gains are modest, and we do not intend to overclaim LED as a uniformly dominant decoding method over every strong baseline on every benchmark. Our main claim is about **generality in a low-headroom regime**: LED is a training-free method that yields **consistent overall improvements** across multiple reasoning benchmarks, model families, and both pass@1 / pass@16, while being compared against **already strong baselines** such as DoLa, ST, and ST-G.
>
> To further substantiate this, we ran 16 independent decoding trials per model and applied a one-sided Wilcoxon signed-rank test. LED's overall improvement over Regular decoding is **statistically significant** across all tested models ($p < 0.01$, one-sided, $n=16$). This confirms that the gains, while modest in absolute terms, are **reliable and not attributable to sampling noise**.
> > W2: RL-induced exploration collapse is suggestive but not fully established
>
> We agree and will soften the wording: RL post-training is a **plausible and empirically supported contributor** to exploration collapse, rather than a formally isolated sole cause. We believe this more accurately reflects the current evidence.
> > W3: computational overhead needs wall-clock / throughput evidence
>
> We agree. Reporting only generation length is insufficient, so we additionally benchmarked wall-clock latency and high-concurrency throughput：
> ### Wall-Clock Benchmarking (batch size=4, 8*A100 GPUs, 16K context)
> |Strategy|Embedding (ms/tok)|Backbone (ms/tok)|LM-Head (ms/tok)|Sampling (ms/tok)|
> |-|-:|-:|-:|-:|
> |Regular | 0.2025 | 5.5674 | 0.0796 | 0.1796 |
> |DoLa | 0.1940 | 5.6129 | 0.1434 | 0.3516 |
> |SoftThinking | 0.2593 | 5.6844 | 0.0919 | 0.4877 |
> |**LED (Ours)**|**0.2049**|**5.6981**|**0.1085**|**0.2363**|
>
> ### High-Concurrency Throughput (batch size=128, 8*H100 GPUs, 16K context)
> | Strategy | Regular | DoLa | SoftThinking | **LED** |
> |-|-:|-:|-:|-:|
> |Througput (tok/s)| 4578 | 3362 | 3025 | **4204** |
> |Relative (%) | 100 | 73.4 | 66.1 | **91.8** |
>
> This supports a more precise claim: LED has **modest runtime overhead** but is more efficient than the strong yet complex decoding baselines.
> > Q1: how is $d$ chosen in the main experiments?
>
> Great question! **We do not search $d$ over benchmarks**. As discussed in Section 4.3 of our manuscript, $d$ is chosen according to the top-$k$ coverage ratio, which is the inherent model attribute. Empirically, performance under different $d$ is not brittle: As illustrated in **Figure 2.1**, increasing $d$ first improves performance, then saturates around $d=8\sim12$. Across most choices of $d$, LED remains above regular decoding.
> > Q2: when do “w/o Exploitation” and “w/ LayerNorm” help or fail?
>
> “w/o Exploitation” mainly hurts when the current token is already highly confident under the final layer. “w/ LayerNorm” tends to make intermediate distributions sharper and can sometimes improve pass@16 by increasing exploration effectiveness over longer reasoning traces, but it may also hurt pass@1 by reducing calibration of the immediate top-1 choice. We will clarify this tradeoff more concretely in the revision.
> > Q3: why maximum entropy for depth selection?
>
> Our key point is that shallow layers are often too immature, while the final layer is already over-collapsed. We therefore construct a depth-wise family of candidate distributions over a shared final-layer top-$k$ candidate set, and use maximum entropy to select the least-collapsed working point among semantically plausible candidates. In this sense, **entropy is used as a practical proxy for recoverable exploration**, not as a correctness guarantee.
> > Q4: sensitivity to top-$k$ & models generality
>
> We agree this matters. We therefore added a top-$k$ sensitivity study over $k=1\sim20$, and the results are shown in **Figure 2.2**. The trend is robust: moving from greedy decoding to a small candidate set already gives clear gains, after which performance stays stable over a broad range. **All $k>1$ outperform regular decoding on both pass@1 and pass@16.**
>
> We do **not** claim LED helps all models. QwQ-32B (**Table 3.1**) is a counter-example: its entropy profile lacks the "intermediate reservoir + final-layer collapse" pattern, and all decoding methods fail to improve its pass@16. LED is most effective when intermediate layers remain informative while the final layer is over-concentrated.
>
> ---
>
> Thank you again for your insightful comments! We will add a Limitations section: (1) absolute gains are modest though statistically significant; (2) some benchmark/model entries are flat or negative; (3) earlier models like QwQ-32B benefit less. We will also incorporate these clarifications, the new latency/throughput results, and the additional sensitivity analyses in the revision.

---

> > ### Author Rebuttal · Reviewer_PVJC · 2026-04-03
> >
> > Thank you for your rebuttal.
> >
> > Most of my concerns are addressed and i have slightly raised my score.

---

> > > ### Author Response · Authors · 2026-04-03
> > >
> > > Thank you for raising your score! We are happy that our response addressed your main concern, and we will incorporate the discussion in our final revision.
> > >
> > > Authors of paper #3484

---

### Official Review · Reviewer_vF2J · 2026-03-13

**Soundness:** 3
**Presentation:** 4
**Significance:** 4
**Originality:** 4
**Overall Recommendation:** 6
**Confidence:** 4

**Summary:**

This paper studies a phenomenon in RL post-trained large reasoning models (LRMs): exploration collapse during decoding. The authors observe that reinforcement learning improves pass@1 accuracy but causes the final-layer token distribution to become overly confident (low entropy), which prevents temperature-based sampling from improving pass@n performance. Interestingly, intermediate transformer layers still retain higher entropy, suggesting that exploration capability remains in the model’s latent representations.

To address this issue, the paper proposes Latent Exploration Decoding (LED), a training-free decoding strategy that leverages intermediate-layer posteriors. LED aggregates latent posteriors from several upper layers, applies top-k filtering based on the final-layer candidates, and selects the distribution with maximum entropy to enable exploration when model confidence is low. The method dynamically balances exploration and exploitation and applies exploration primarily during the reasoning (DeepThink) phase.

**Compliance With Llm Reviewing Policy:**

Affirmed.

**Key Questions For Authors:**

1. LED relies on entropy-based selection of aggregated latent posteriors. How sensitive is the method to the choice of exploration depth \(d\) and top-k filtering? A more systematic sensitivity analysis would help assess robustness and practical usability.


2. The method uses intermediate layer logits via the LM head (early-exit decoding). How does LED behave for models whose intermediate representations are less aligned with the LM head (e.g., architectures with different normalization or head-sharing designs)? Clarifying this could help understand the generality of the approach.

**Limitations:**

Yes.

**Strengths And Weaknesses:**

**Soundness**
The paper is generally technically sound. The authors support their claims with empirical analysis showing entropy collapse in the final layer and higher entropy in intermediate layers. Experiments across several models and benchmarks demonstrate consistent improvements in pass@1 and pass@16. However, the theoretical explanation of entropy collapse is mostly intuitive and the empirical gains are relatively modest.

**Presentation**
The paper is clearly written and well structured. The motivation, method, and experiments are logically organized, and figures help explain the key idea. Minor improvements could include simplifying some technical descriptions and clarifying implementation details for reproducibility.

**Significance**
The work addresses an important issue in reasoning LLMs: reduced exploration after RL post-training. A training-free decoding method that restores exploration has practical value, especially for reasoning and code-generation tasks, although the improvements are incremental.

**Originality**
The work provides moderate novelty. While prior work has used intermediate layers during decoding, the paper introduces a new perspective by exploiting entropy differences across layers and selecting high-entropy latent distributions for exploration.

---

> ### Author Rebuttal · Authors · 2026-03-30
>
> Dear Reviewer #vF2J,
>
> Thank you for your positive assessment of the paper and for the two concrete questions on robustness and generality. We address them below. **[All figures are provided in the anonymous webpage](https://anonymous.4open.science/r/LED-Rebuttal/README.md)**.
>
> ---
>
> > Q1: How sensitive is the method to the choice of exploration depth d and top-$k$ filtering?
>
> We agree that robustness to $d$ and top-$k$ is important for practical usability, and we have added a more systematic sensitivity analysis.
>
> For the exploration depth $d$, we already included an ablation in the original manuscript (and also highlighted in **Figure 2.1**). Across pass@1 to pass@16, the trend is highly consistent: increasing $d$ from 1 first brings clear gains, then the performance saturates around $d=8\sim12$, and only slightly decreases when d becomes too large. Importantly, **across most choices of $d$, LED remains above regular decoding (the horizontal dashed lines)**. This indicates that LED is not narrowly tuned to one specific depth, and that its benefit is robust as long as the exploration reaches sufficiently informative intermediate layers.
>
> For top-$k$, we additionally tested $k \in \{1,4,8,12,16,20\}$, and the results are shown in [Figure 2.2](https://anonymous.4open.science/r/LED-Rebuttal/README.md). The results show the same pattern: moving from greedy decoding ($k=1$) to a small candidate set already yields a substantial gain; performance then stays stable over a broad range of $k$. In particular, pass@1 improves and remains around 79\% once $k>4$, while pass@16 increases further and peaks around $k=12\sim16$, with only a small drop at $k=20$. **All $k>1$ settings outperform regular decoding on both pass@1 and pass@16**.
>
> Overall, these results suggest that **LED is not brittle to either $d$ or top-$k$**, and works well over a reasonably wide parameter range. This robustness is consistent with the design of LED: it does not rely on one exact best layer, but on the existence of a reasonably broad intermediate regime that remains informative yet not fully collapsed.
>
> > Q2: How does LED behave for models whose intermediate representations are less aligned with the LM head?
>
> This is an important question, and our current evidence suggests that LED indeed depends on a meaningful degree of alignment between intermediate representations and the shared LM head. More precisely, LED is most effective when the model exhibits the entropy pattern that motivates our method: the final layer becomes over-sharpened, while a band of intermediate layers still preserves useful uncertainty over plausible candidates.
>
> A useful counter-example is QwQ-32B, one of the earliest open-source reasoning models. As shown in our appendix (also shown in **Figure 3.1** and **Table 3.1**), QwQ-32B behaves differently from the other model families: its layerwise entropy profile is much less consistent with the “intermediate entropy reservoir + final-layer collapse” pattern seen in Qwen3-Thinking, DeepSeek-R1-Distill, and MiMo-style models. Correspondingly, all decoding methods become less effective on pass@16 for QwQ-32B, and LED no longer provides the same level of exploration gain as on newer post-trained reasoning models. Nevertheless, LED still gives the best pass@1 among the compared decoding methods on this model.
>
> We will clarify this boundary in the revision: LED is not intended to assume that every architecture has equally well-calibrated intermediate logits. Rather, it is a decoding method tailored to the increasingly common class of post-trained reasoning LLMs whose intermediate layers remain semantically meaningful while the final layer becomes overly concentrated. Our current cross-model results suggest that this condition holds for modern reasoning models, but not uniformly for earlier ones such as QwQ-32B.
>
> ---
>
> Thank you again for your positive assessment and thoughtful questions! We will incorporate these additional analyses and clarifications in the revision, and we are happy to provide additional clarification if needed.

---

> > ### Author Rebuttal · Reviewer_vF2J · 2026-03-31
> >
> > Thank you for addressing the issue I concern about. I really enjoy your paper and hope you can open-source the code as soon as possible. I will keep the score.

---

> > > ### Author Response · Authors · 2026-04-02
> > >
> > > We are glad the additional analyses resolved your concerns! And we sincerely appreciate your interest in the work.
> > >
> > > We will incorporate additional analyses and clarifications in the revision, and open-source the code after this paper is accepted.
> > >
> > > Authors of paper #3484

---

### Decision · Program_Chairs · 2026-04-30

**Decision:**

Accept (regular)

**Comment:**

The paper documents an interesting phenomenon: RL post-training sharply reduces final-layer entropy in reasoning models, rendering temperature-based sampling ineffective for generating diverse outputs. The proposed method, LED, restores diversity by aggregating intermediate-layer posteriors — a simple, training-free approach. The rebuttal strengthened the paper considerably by showing that LED also improves RL-rollout quality (not just inference-time search) and by providing throughput comparisons against DoLa and SoftThinking, where LED is both faster and more accurate. The remaining concern from one reviewer is that the aggregation rule is heuristic and the absolute gains are modest. This is a valid limitation, but the improvements are statistically significant and consistent across models and benchmarks. The paper identifies a real problem and proposes a practical solution with clear empirical support. Recommended for acceptance.